# FILTERED-COPHY: UNSUPERVISED LEARNING OF COUNTERFACTUAL PHYSICS IN PIXEL SPACE

**Steeven Janny**
LIRIS, INSA Lyon, France
steeven.janny@insa-lyon.fr

**Fabien Baradel**
Naver Labs Europe, France
fabien.baradel@naverlabs.com

**Natalia Neverova**
Meta AI
nneverova@fb.com

**Madiha Nadri**
LAGEPP, Univ. Lyon 1, France
madiha.nadri-wolf@univ-lyon1.fr

**Greg Mori**
Simon Fraser Univ., Canada
mori@cs.sfu.ca

**Christian Wolf**
LIRIS, INSA Lyon, France
christian.wolf@insa-lyon.fr

## ABSTRACT

Learning causal relationships in high-dimensional data (images, videos) is a hard task, as they are often defined on low-dimensional manifolds and must be extracted from complex signals dominated by appearance, lighting, textures and also spurious correlations in the data. We present a method for learning counterfactual reasoning of physical processes in pixel space, which requires the prediction of the impact of interventions on initial conditions. Going beyond the identification of structural relationships, we deal with the challenging problem of forecasting raw video over long horizons. Our method does not require the knowledge or supervision of any ground truth positions or other object or scene properties. Our model learns and acts on a suitable hybrid latent representation based on a combination of dense features, sets of 2D keypoints and an additional latent vector per keypoint. We show that this better captures the dynamics of physical processes than purely dense or sparse representations. We introduce a new challenging and carefully designed counterfactual benchmark for predictions in pixel space and outperform strong baselines in physics-inspired ML and video prediction.

## 1 INTRODUCTION

Reasoning on complex, multi-modal and high-dimensional data is a natural ability of humans and other intelligent agents (Martin-Ordas et al., 2008), and one of the most important and difficult challenges of AI. While machine learning is well suited for capturing regularities in high-dimensional signals, in particular by using high-capacity deep networks, some applications also require an accurate modeling of causal relationships. This is particularly relevant in physics, where causation is considered as a fundamental axiom. In the context of machine learning, correctly capturing or modeling causal relationships can also lead to more robust predictions, in particular better generalization to out-of-distribution samples, indicating that a model has overcome the exploitation of biases and shortcuts in the training data. In recent literature on physics-inspired machine learning, causality has often been forced through the addition of prior knowledge about the physical laws that govern the studied phenomena, e.g. (Yin et al., 2021). A similar idea lies behind structured causal models, widely used in the causal inference community, where domain experts model these relationships directly in a graphical notation. This particular line of work allows to perform predictions beyond statistical forecasting, for instance by predicting unobserved counterfactuals, the impact of unobserved interventions (Balke & Pearl, 1994) — "*What alternative outcome would have happened, if the observed event X had been replaced with an event Y (after an intervention)*". Counterfactuals are interesting, as causality intervenes through the effective modification of an outcome. As an example, taken from (Schölkopf et al., 2021), an agent can identify the direction of a causal relationship between an umbrella and rain from the fact that removing an umbrella will not affect the weather.

We focus on counterfactual reasoning on high-dimensional signals, in particular videos of complex physical processes. Learning such causal interactions from data is a challenging task, as spurious correlations are naturally and easily picked up by trained models. Previous work in this direction

was restricted to discrete outcomes, as in *CLEVRER* (Yi et al., 2020), or to the prediction of 3D trajectories, as in *CoPhy* (Baradel et al., 2020), which also requires supervision of object positions. In this work, we address the hard problem of predicting the alternative (counterfactual) outcomes of physical processes in pixel space, i.e. we forecast sequences of 2D projective views of the 3D scene, requiring the prediction over long horizons (150 frames corresponding to $\sim$ 6 seconds). We conjecture that causal relationships can be modeled on a low dimensional manifold of the data, and propose a suitable latent representation for the causal model, in particular for the estimation of the confounders and the dynamic model itself. Similar to V-CDN (Kulkarni et al., 2019; Li et al., 2020), our latent representation is based on the unsupervised discovery of keypoints, complemented by additional information in our case. Indeed, while keypoint-based representations can easily be encoded from visual input, as stable mappings from images to points arise naturally, we claim that they are not the most suitable representation for dynamic models. We identified and addressed two principal problems: (i) the individual points of a given set are discriminated through their 2D positions only, therefore shape, geometry and relationships between multiple moving objects need to be encoded through the relative positions of points to each other, and (ii) the optimal representation for a physical dynamic model is not necessarily a 2D keypoint space, where the underlying object dynamics has also been subject to the imaging process (projective geometry).

We propose a new counterfactual model, which learns a sparse representation of visual input in the form of 2D keypoints coupled with a (small) set of coefficients per point modeling complementary shape and appearance information. Confounders (object masses and initial velocities) in the studied problem are extracted from this representation, and a learned dynamic model forecasts the entire trajectory of these keypoints from a single (counterfactual) observation. Building on recent work in data-driven analysis of dynamic systems (Janny et al., 2021; Peralez & Nadri, 2021), the dynamic model is presented in a higher-dimensional state space, where dynamics are less complex. We show, that these design choices are key to the performance of our model, and that they significantly improve the capability to perform long-term predictions. Our proposed model outperforms strong baselines for physics-informed learning of video prediction.

We introduce a new challenging dataset for this problem, which builds on *CoPhy*, a recent counterfactual physics benchmark (Baradel et al., 2020). We go beyond the prediction of sequences of 3D positions and propose a counterfactual task for predictions in pixel space after interventions on initial conditions (displacing, re-orienting or removing objects). In contrast to the literature, our benchmark also better controls for the identifiability of causal relationships and counterfactual variables and provides more accurate physics simulation.

## 2 RELATED WORK

**Counterfactual (CF) reasoning** — and learning of causal relationships in ML was made popular by works of J. Pearl, e.g. (Pearl, 2000), which motivate and introduce mathematical tools detailing the principles of *do-calculus*, i.e. study of unobserved interventions on data. A more recent survey links these concepts to the literature in ML (Schölkopf et al., 2021). The last years have seen the emergence of several benchmarks for CF reasoning in physics. CLEVRER (Yi et al., 2020) is a visual question answering dataset, where an agent is required to answer a CF question after observing a video showing 3D objects moving and colliding. Li et al. (2020) introduce a CF benchmark with two tasks: a scenario where balls interact with each other according to unknown interaction laws (such as gravity or elasticity), and a scenario where clothes are folded by the wind. The agent needs to identify CF variables and causal relationships between objects, and to predict future frames. Co-Phy (Baradel et al., 2020) clearly dissociates the observed experiment from the CF one, and contains three complex 3D scenarios involving rigid body dynamics. However, the proposed method relies on the supervision of 3D object positions, while our work does not require any meta data.

**Physics-inspired ML** — and learning visual dynamics has been dealt early on with recurrent models (Srivastava et al., 2015; Finn et al., 2016; Lu et al., 2017), or GANs (Vondrick et al., 2016; Mathieu et al., 2016). Kwon & Park (2019) adopt a Cycle-GAN with two discriminator heads, in charge of identifying false images and false sequences in order to improve the temporal consistency of the model in long term prediction. Nonetheless, the integration of causal reasoning and prior knowledge in these models is not straightforward. Typical work in physics-informed models relies on disentanglement between physics-informed features and residual features (Villegas et al., 2017a;

Denton & Birodkar, 2017) and may incorporate additional information based on the available priors on the scene (Villegas et al., 2017b; Walker et al., 2017). PhyDNet Le Guen & Thome (2020) explicitly disentangles visual features from dynamical features, which are supposed to follow a PDE. It achieves SOTA performance on Human3.6M (Ionescu et al., 2014) and Sea Surface Temperature (de Bezenac et al., 2018), but we show that it fails on our challenging benchmark.

**Keypoint detection** — is a well researched problem in vision with widely used handcrafted baselines (Lowe, 1999). New unsupervised variants emerged recently and have been shown to provide a suitable object-centric representation, close to attention models, which simplify the use of physical and/or geometric priors (Locatello et al., 2020; Veerapaneni et al., 2020). They are of interest in robotics and reinforcement learning, where a physical agent has to interact with objects (Kulkarni et al., 2019; Manuelli et al., 2020; 2019). KeypointsNet (Suwajanakorn et al., 2018) is a geometric reasoning framework, which discovers meaningful keypoints in 3D through spatial coherence between viewpoints. Close to our work, (Minderer et al., 2019) proposes to learn a keypoints-based stochastic dynamic model. However, the model is not suited for CF reasoning in physics and may suffer from inconsistency in the prediction of dynamics over long horizons.

## 3 THE **FILTERED-COPHY** BENCHMARK

We build on *CoPhy* (Baradel et al., 2020), retaining its strengths, but explicitly focusing on a counterfactual scenario in pixel space and eliminating the ill-posedness of tasks we identified in the existing work. Each data sample is called an *experiment*, represented as a pair of trajectories: an *observed* one with initial condition $X_0 = \mathbf{A}$ and outcome $X_{t=1..T} = \mathbf{B}$ (a sequence), and a *counterfactual* one $\bar{X}_0 = \mathbf{C}$ and $\bar{X}_{t=1..T} = \mathbf{D}$ (a sequence). Throughout this paper we will use the letters $\mathbf{A}, \mathbf{B}, \mathbf{C}$ and $\mathbf{D}$ to distinguish the different parts of each experiment. The initial conditions $\mathbf{A}$ and $\mathbf{C}$ are linked through a *do-operator* $do(X_0 = \mathbf{C})$, which modifies the initial condition (Pearl, 2018). Experiments are parameterized by a set of intrinsic physical parameters $z$ which are not observable from a single initial image $\mathbf{A}$. We refer to these as *confounders*. As in *CoPhy*, in our benchmark the do-operator is observed during training, but confounders are not — they have been used to generate the data, but are not used during training or testing. Following (Pearl, 2018), the counterfactual task consists in inferring the counterfactual outcome $\mathbf{D}$ given the observed trajectory $\mathbf{AB}$ and the counterfactual initial state $\mathbf{C}$, following a three-step process:

①  **Abduction**: use the observed data $\mathbf{AB}$ to compute the counterfactual variables, i.e. physical parameters, which are not affected by the do-operation.

②  **Action**: update the causal model; keep the same identified confounders and apply the do-operator, i.e. replace the initial state $\mathbf{A}$ by $\mathbf{C}$.

③  **Prediction** : Compute the counterfactual outcome $\mathbf{D}$ using the causal graph.

The benchmark contains three scenarios involving rigid body dynamics. `BlocktowerCF` studies stable and unstable 3D cube towers, the confounders are masses. `BallsCF` focuses on 2D collisions between moving spheres (confounders are masses and initial velocities). `CollisionCF` is about collisions between a sphere and a cylinder (confounders are masses and initial velocities) (Fig. 1).

Unlike *CoPhy*, our benchmark involves predictions in RGB pixel space only. The do-operation consists in visually observable interventions on $\mathbf{A}$, such as moving or removing an object. The confounders cannot be identified from the single-frame observation $\mathbf{A}$, identification requires the analysis of the entire $\mathbf{AB}$ trajectory.

**Identifiability of confounders** — For an experiment $(\mathbf{AB}, \mathbf{CD}, z)$ to be well-posed, the confounders $z$ must be retrievable from $\mathbf{AB}$. For example, since the masses of a stable cube tower cannot be identified generally in all situations, it can be impossible to predict the counterfactual outcome of an unstable tower, as collisions are not resolvable without known masses. In contrast to *CoPhy*, we ensure that each experiment $\psi : (X_0, z) \mapsto X_{t=1..T}$, given initial condition $X_0$ and confounders $z$, is well posed and satisfies the following constraints:

**Definition 1** *(Identifiability, (Pearl, 2018)) The experiment* $(\mathbf{AB}, \mathbf{CD}, z)$ *is identifiable if, for any set of confounders* $z'$:

$$\psi(\mathbf{A}, z) = \psi(\mathbf{A}, z') \Rightarrow \psi(\mathbf{C}, z) = \psi(\mathbf{C}, z'). \tag{1}$$

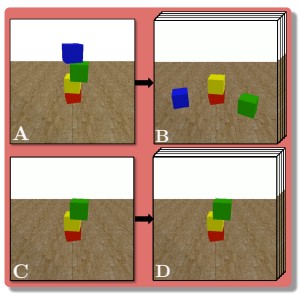 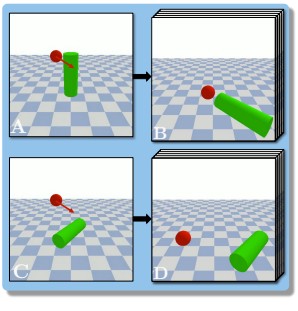 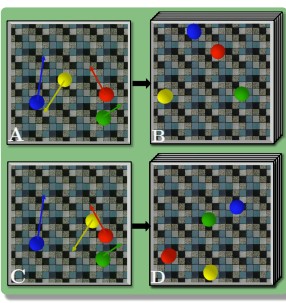

(a) BlocktowerCF (BT-CF)  (b) CollisionCF (C-CF)  (c) BallsCF (B-CF)

Figure 1: The Filtered-CoPhy benchmark suite contains three challenging scenarios involving 2D or 3D rigid body dynamics with complex interactions, including collision and resting contact. Initial conditions **A** are modified to **C** by an intervention. Initial motion is indicated through arrows.

In an identifiable experiment there is no pair $(z, z')$ that gives the same trajectory **AB** but different counterfactual outcomes **CD**. Details on implementation and impact are in appendix A.1.

**Counterfactuality** — We enforce sufficient difficulty of the problem through the meaningfulness of confounders. We remove initial situations where the choice of confounder values has no significant impact on the final outcome:

**Definition 2** (Counterfactuality). *Let $z^k$ be the set of confounders $z$, where the $k^{th}$ value has been modified. The experiment $(\mathbf{AB}, \mathbf{CD}, z)$ is counterfactual if and only if:*

$$\exists k : \psi(\mathbf{C}, z^k) \neq \psi(\mathbf{C}, z). \tag{2}$$

In other words, we impose the existence of an object of the scene for which the (unobserved) physical properties have a determining effect on the trajectory. Details on how this constraint was enforced are given in appendix A.2.

**Temporal resolution** — the physical laws we target involve highly non-linear phenomena, in particular collision and resting contacts. Collisions are difficult to learn because their actions are both intense, brief, and highly non-linear, depending on the geometry of the objects in 3D space. The temporal resolution of physical simulations is of prime importance. A parallel can be made with Nyquist-Shannon frequency, as a

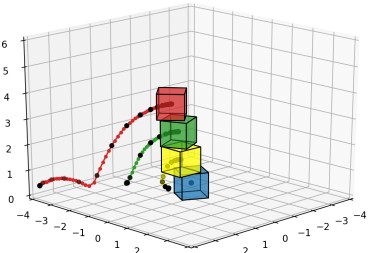

Figure 2: Impact of temporal frequency on dynamics, 3D trajectories of each cube are shown. Black dots are sampled at 5 FPS, colored dots at 25 FPS. Collisions between the red cube and the ground are not well described by the black dots, making it hard to infer physical laws from regularities in data.

trajectory sampled with too low frequency cannot be reconstructed with precision. We simulate and record trajectories at 25 FPS, compared to 5 FPS chosen in *CoPhy*, justified with two experiments. Firstly, Fig. 2 shows the trajectories of the center of masses of cubes in BlocktowerCF, colored dots are shown at 25 FPS and black dots at 5 FPS. We can see that collisions with the ground fall below the sampling rate of 5 FPS, making it hard to infer physical laws from regularities in data at this frequency. A second experiment involves learning a prediction model at different frequencies, confirming the choice 25 FPS — details are given in appendix A.3.

## 4 UNSUPERVISED LEARNING OF COUNTERFACTUAL PHYSICS

We introduce a new model for counterfactual learning of physical processes capable of predicting visual sequences **D** in the image space over long horizons. The method does not require any supervision other than videos of observed and counterfactual experiences. The code is publicly available online at **https://filteredcophy.github.io**. The model consists of three parts, learning the latent representation and its (counterfactual) dynamics:

- **The encoder (De-Rendering module)** learns a hybrid representation of an image in the form of a (i) dense feature map and (ii) 2D keypoints combined with (iii) a low-dimensional vector of coef-

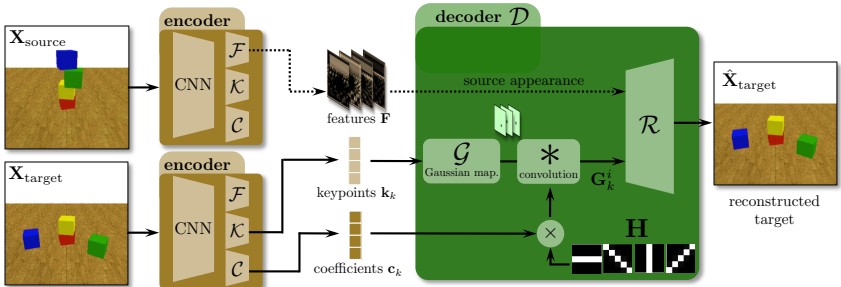

Figure 3: We de-render visual input into a latent space composed of a dense feature map $\mathbf{F}$ modeling static information, a set of keypoints $\mathbf{k}$, and associated coefficients $\mathbf{c}$. We here show the training configuration taking as input pairs $(\mathbf{X}_{\text{source}}, \mathbf{X}_{\text{target}})$ of images. Without any supervision, a tracking strategy emerges naturally through the unsupervised objective: we optimize reconstruction of $\mathbf{X}_{\text{target}}$, given features from $\mathbf{X}_{\text{source}}$ and keypoints+coefficients from $\mathbf{X}_{\text{target}}$.

ficients, see Fig. 3. Without any state supervision we show that the model learns a representation which encodes positions in keypoints and appearance and orientation in the coefficients.

- **The Counterfactual Dynamic (CoDy) model** based on recurrent graph networks, in the lines of (Baradel et al., 2020). It estimates a latent representation of the confounders $z$ from the keypoint + coefficient trajectories of $\mathbf{AB}$ provided by the encoder, and then predicts $\mathbf{D}$ in this same space.

- **The decoder** that uses the predicted keypoints to generate a pixel-space representation of $\mathbf{D}$.

## 4.1 DISENTANGLING VISUAL INFORMATION FROM DYNAMICS

The encoder takes an input image $\mathbf{X}$ and predicts a representation with three streams, sharing a common conv. backbone, as shown in Fig. 3. We propose an unsupervised training objective, which favors the emergence of a latent representation disentangling static and dynamic information.

1. A dense feature map $\mathbf{F} = \mathcal{F}(\mathbf{X})$, which contains static information, such as the background.

2. A set of 2D keypoints $\mathbf{k} = \mathcal{K}(\mathbf{X})$, which carry positional information from moving objects.

3. A set of corresponding coefficients $\mathbf{c} = \mathcal{C}(\mathbf{X})$, one vector $\mathbf{c}_k$ of size $C$ per keypoint $k$, which encodes orientation and appearance information.

The unsupervised objective is formulated on pairs of images $(\mathbf{X}_{\text{source}}, \mathbf{X}_{\text{target}})$ randomly sampled from same $\mathbf{D}$ sequences (see appendix D.1 for details on sampling). Exploiting an assumption on the absence of camera motion[1], the goal is to favor the emergence of disentangling static and dynamic information. To this end, both images are encoded, and the reconstruction of the target image is predicted with a decoder $\mathcal{D}$ fusing the source dense feature map and the target keypoints and coefficients. This formulation requires the decoder to aggregate dense information from the source and sparse values from the target, naturally leading to motion being predicted by the latter.

On the decoder $\mathcal{D}$ side, we add inductive bias, which favors the usage of the 2D keypoint information in a spatial way. The 2D coordinates $\mathbf{k}_k$ for each keypoint $k$ are encoded as Gaussian heatmaps $\mathcal{G}(\mathbf{k}_k)$, i.e. 2D Gaussian functions centered on the keypoint position. The additional coefficient information, carrying appearance information, is then used to deform the Gaussian mapping into an anisotropic shape using a fixed filter bank $\mathbf{H}$, as follows:

$$\mathcal{D}(\mathbf{F}, \mathbf{k}, \mathbf{c}) = \mathcal{R}\left(\mathbf{F}, \mathbf{G}_1^1, ..., \mathbf{G}_1^C, \mathbf{G}_2^1, ..., \mathbf{G}_K^C\right), \quad \mathbf{G}_k^i = \mathbf{c}_k^{C+1} \mathbf{c}_k^i \left(\mathcal{G}(\mathbf{k}_k) * \mathbf{H}_i\right), \qquad (3)$$

where $\mathcal{R}(...)$ is a refinement network performing trained upsampling with transposed convolutions, whose inputs are stacked channelwise. $\mathbf{G}_k^i$ are Gaussian mappings produced from keypoint positions $\mathbf{k}$, deformed by filters from bank $\mathbf{H}$ and weighted by coefficients $\mathbf{c}_k^i$. The filters $\mathbf{H}_i$ are defined as fixed horizontal, vertical and diagonal convolution kernels. This choice is discussed in section 5. The joint encoding and decoding pipeline is illustrated in Fig. 3.

---

[1]If this assumption is not satisfied, global camera motion could be compensated after estimation.

The model is trained to minimize the mean squared error (MSE) reconstruction loss, regularized with a loss on spatial gradients $\nabla \mathbf{X}$ weighted by hyper-parameters $\gamma_1, \gamma_2 \in \mathbb{R}$:

$$\mathcal{L}_{\text{deren}} = \gamma_1 \left\| \mathbf{X}_{\text{target}} - \hat{\mathbf{X}}_{\text{target}} \right\|_2^2 + \gamma_2 \left\| \nabla \mathbf{X}_{\text{target}} - \nabla \hat{\mathbf{X}}_{\text{target}} \right\|_2^2, \tag{4}$$

where $\hat{X}_{\text{target}} = \mathcal{D}(\mathcal{F}(\mathbf{X}_{\text{source}}), \mathcal{K}(\mathbf{X}_{\text{target}}), \mathcal{C}(\mathbf{X}_{\text{target}}))$ is the reconstructed image, $\gamma_1, \gamma_2$ are weights.

**Related work** – our unsupervised objective is somewhat related to *Transporter* (Kulkarni et al., 2019), which, as our model, computes visual feature vectors $F_{\text{source}}$ and $F_{\text{target}}$ as well as 2D keypoints $K_{\text{source}}$ and $K_{\text{target}}$, modeled as a 2D vector via Gaussian mapping. It leverages a handcrafted transport equation: $\hat{\Psi}_{\text{target}} = F_{\text{source}} \times (1 - K_{\text{source}}) \times (1 - K_{\text{target}}) + F_{\text{target}} \times K_{\text{target}}$. As in our case, the target image is reconstructed through a refiner network $\hat{X}_{\text{target}} = \mathcal{R}(\hat{\Psi}_{\text{target}})$. The transporter suffers from a major drawback when used for video prediction, as it requires parts of the target image to reconstruct the target image — the model was originally proposed in the context of RL and control, where reconstruction is not an objective. It also does not use shape coefficients, requiring shapes to be encoded by several keypoints, or abusively be carried through the dense features $F_{\text{target}}$. This typically leads to complex dynamics non representative of the dynamical objects. We conducted an in-depth comparison between the Transporter and our representation in appendix C.2.

## 4.2 Dynamic model and confounder estimation

Our counterfactual dynamic model (CoDy) leverages multiple graph network (GN) based modules (Battaglia et al., 2016) that join forces to solve the counterfactual forecasting tasks of **Filtered-CoPhy**. Each one of these networks is a classical GN, abbreviated as $\mathcal{GN}(\mathbf{x}_k)$, which contextualizes input node embeddings $\mathbf{x}_k$ through incoming edge interactions $\mathbf{e}_{ik}$, providing output node embeddings $\hat{\mathbf{x}}_k$ (parameters are not shared over the instances):

$$\mathcal{GN}(\mathbf{x}_k) = \hat{\mathbf{x}}_k, \text{ such that } \hat{\mathbf{x}}_k = g\left(\mathbf{x}_k, \sum_i \mathbf{e}_{ik}\right) \text{ with } \mathbf{e}_{ij} = f(\mathbf{x}_i, \mathbf{x}_j), \tag{5}$$

where $f$ is a message-passing function and $g$ is an aggregation function.

We define the state of frame $\mathbf{X}_t$ at time $t$ as a stacked vector composed of keypoints and coefficients computed by the encoder (the de-rendering module), i.e. $\mathbf{s}(t) = [\mathbf{s}_1(t) \ ... \ \mathbf{s}_k(t)]$ where $\mathbf{s}_k(t) = [\mathbf{k}_k \ \mathbf{c}_k^1 ... \mathbf{c}_k^{C+1}](t)$. In the lines of (Baradel et al., 2020), given the original initial condition and outcome $\mathbf{AB}$, CoDy estimates an unsupervised representation $\mathbf{u}_k$ of the latent confounder variables per keypoint $k$ through the counterfactual estimator (*CF estimator* in Fig. 4). It first contextualizes the sequence $\mathbf{s}^{\mathbf{AB}}(t)$ through a graph network $\mathcal{GN}(\mathbf{s}^{\mathbf{AB}}(t)) = \mathbf{h}^{\mathbf{AB}}(t) = [\mathbf{h}_1(t) \ ... \ \mathbf{h}_K(t)]$. We then model the temporal evolution of this representation with a gated recurrent unit (Cho et al., 2014) per keypoint, sharing parameters over keypoints, taking as input the sequence $\mathbf{h}_k$. Its last hidden vector is taken as the confounder estimate $\mathbf{u}_k$.

Recent works on the Koopman Operator (Lusch et al., 2018) and Kazantzis-Kravaris-Luenberger Observer (Janny et al., 2021; Peralez & Nadri, 2021) have theoretically shown that, under mild assumptions, there exists a latent space of higher dimension, where a dynamical system given as an EDP can have a simpler dynamics. Inspired by this idea, we used an encoder-decoder structure within CoDy, which projects our dynamic system into a higher-dimensional state space, performs forecasting of the dynamics in this latent space, and then projects predictions back to the original keypoint space. Note that this dynamics encoder/decoder is different from the encoder/decoder of the de-rendering/rendering modules discussed in section 4.1. The state encoder $\mathcal{E}(\mathbf{s}(t)) = \boldsymbol{\sigma}(t)$ is modeled as a graph network $\mathcal{GN}$, whose aggregation function projects into an output embedding space $\boldsymbol{\sigma}(t)$ of dimension 256. The decoder $\Delta(\boldsymbol{\sigma}(t)) = \mathbf{s}(t)$ temporally processes the individual contextualized states $\boldsymbol{\sigma}(t)$ with a GRU, followed by new contextualization with a graph network $\mathcal{GN}$. Details on the full architecture are provided in appendix D.2.

The dynamic model CoDy performs forecasting in the higher-dimensional space $\boldsymbol{\sigma}(t)$, computing a displacement vector $\boldsymbol{\delta}(t+1)$ such that $\boldsymbol{\sigma}(t+1) = \boldsymbol{\sigma}(t) + \boldsymbol{\delta}(t+1)$. It takes the projected state embeddings $\sigma_k^{\mathbf{CD}}(t)$ per keypoint $k$ concatenated with the confounder representation $\mathbf{u}_k$ and contextualizes them with a graph network, resulting in embeddings $\mathbf{h}_k^{\mathbf{CD}}(t) = \mathcal{GN}([\boldsymbol{\sigma}_k^{\mathbf{CD}}(t), \mathbf{u}_k])$, which are processed temporally by a GRU. We compute the displacement vector at time $t+1$

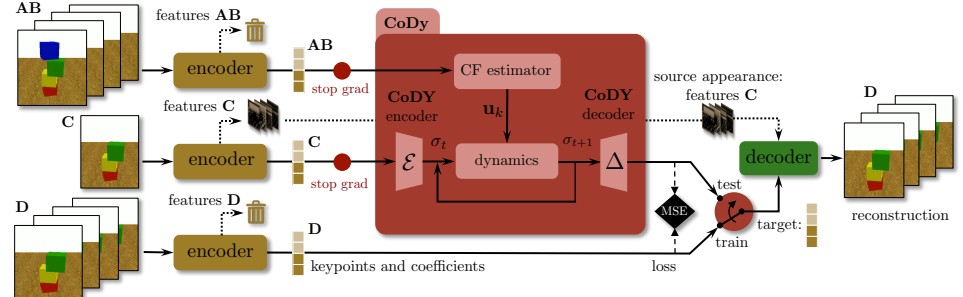

Figure 4: During training, we disconnect the dynamic prediction module (CoDy) from the rendering module (decoder). On test time, we reconnect the two modules. CoDy forecasts the counterfactual outcome **D** from the sparse keypoints representation of **AB** and **C**. The confounders are discovered in an unsupervised manner and provided to the dynamical model.

|  |  | **Ours** | | UV-CDN | | PhyD | Pred |
|---|---|---|---|---|---|---|---|
|  |  | N | 2N | N | 2N | NET | RNN |
| BT-CF | PSNR | 23.48 | **24.69** | 21.07 | 21.99 | 16.49 | 22.04 |
|  | L-PSNR | 25.51 | **26.79** | 22.36 | 23.64 | 23.03 | 24.97 |
| B-CF | PSNR | 21.19 | **21.33** | 19.51 | 19.54 | 18.56 | 22.31 |
|  | L-PSNR | **23.88** | 24.12 | 22.35 | 22.38 | 22.55 | 22.63 |
| C-CF | PSNR | 24.09 | 24.09 | 23.73 | 23.83 | 19.69 | **24.70** |
|  | L-PSNR | 26.07 | **26.55** | 26.08 | 26.34 | 24.61 | 26.39 |

|  | Copy B | Copy C | **Ours** |
|---|---|---|---|
| BT-CF | 43.2 | 20.0 | **9.58** |
| B-CF | 44.3 | 92.3 | **36.12** |
| C-CF | 7.6 | 40.3 | **5.14** |

Copy B: assumes absence of intervention
(always outputs sequence **B**).
Copy C: assumes that the tower is stable
(always outputs input **C**).

Table 1: Comparison with the state-of-the-art models in physics-inspired machine learning of video signals, reporting reconstruction error (PSNR and introduced L-PSNR).

Table 2: Comparison with copying baselines. We report MSE$\times 10^{-3}$ on prediction of keypoints + coefficients.

as a linear transformation from the hidden state of the GRU. We apply the dynamic model in an auto-regressive way to forecast long-term trajectories in the projected space $\sigma_k$, and apply the state decoder to obtain a prediction $\hat{\mathbf{s}}^{\mathbf{CD}}(t)$.

The dynamic model is trained with a loss in keypoint space,

$$\mathcal{L}_{\text{physics}} = \sum_t \|\mathbf{s}^{\mathbf{CD}}(t) - \hat{\mathbf{s}}^{\mathbf{CD}}(t)\|_2^2 + \gamma_3 \|\mathbf{s}^{\mathbf{CD}}(t) - \Delta(\mathcal{E}(\mathbf{s}^{\mathbf{CD}}(t)))\|_2^2. \tag{6}$$

The first term enforces the model to learn to predict the outcomes and the second term favors correct reconstruction of the state in keypoint space. The terms are weighted with a scalar parameter $\gamma_3$.

## 4.3 TRAINING

End-to-end training of all three modules jointly is challenging, as the same pipeline controls both the keypoint-based state representation and the dynamic module (CoDy), involving two adversarial objectives: optimizing reconstruction pushes the keypoint encoding to be as representative as possible, but learning the dynamics favors a simple representation. Faced with these two contradictory tasks, the model is numerically unstable and rapidly converges to regression to the mean. As described above, we separately train the encoder+decoder pair without dynamic information on reconstruction only, c.f. Equation (4). Then we freeze the parameters of the keypoint detector and train CoDy to forecast the keypoints from **D** minimizing the loss in Equation (6).

## 5 EXPERIMENTS

We compare the proposed model to three strong baselines for physics-inspired video prediction.

- **PhyDNet** (Le Guen & Thome, 2020) is a non-counterfactual video prediction model that forecasts future frames using a decomposition between (a) a feature vector that temporally evolves via an LSTM and (b) a dynamic state that follows a PDE learned through specifically designed cells.

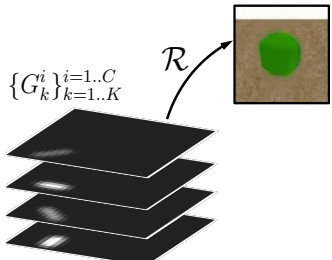

$\{G_k^i\}_{k=1..K}^{i=1..C}$

$\mathcal{R}$

Figure 5: Network $\mathcal{R}$ learns a distortion from multiple oriented ellipses to target shapes.

| # Coefficients: | | ✓ | | | ✗ | |
| # Keypoints: | N | 2N | 4N | N | 2N | 4N |
|---|---|---|---|---|---|---|
| **BT-CF** PSNR | 23.48 | **24.69** | 23.54 | 22.71 | 23.28 | 23.17 |
| L-PSNR | 21.75 | **23.03** | 21.80 | 21.18 | 21.86 | 21.70 |
| **B-CF** PSNR | 21.19 | 21.33 | **21.37** | 20.49 | 21.09 | 20.97 |
| L-PSNR | **27.88** | 27.16 | 27.07 | 26.33 | 27.07 | 26.73 |
| **C-CF** PSNR | 24.09 | 24.09 | **24.26** | 23.84 | 23.66 | 24.06 |
| L-PSNR | 23.32 | **23.46** | 23.44 | 22.58 | 22.81 | 23.45 |

Table 3: Impact of having additional orientation/shape coefficients (✓) compared to the keypoint-only solution (✗), for different numbers of keypoints: equal to number of objects (= N), 2N and 4N.

- **V-CDN** (Li et al., 2020) is a counterfactual model based on keypoints, close to our work. It identifies confounders from the beginning of a sequence and learns a keypoint predictor through auto-encoding using the Transporter equation (see discussion in Sect. 4.1). As it is, it cannot be used for video prediction and is incomparable with our work, see details in appendix C. We therefore replace the Transporter by our own de-rendering/rendering modules, from which we remove the additional coefficients. We refer to this model as UV-CDN (for Unsupervised V-CDN).

- **PredRNN** (Wang et al., 2017) is a ConvLSTM-based video prediction model that leverages spatial and temporal memories a through a spatiotemporal LSTM cell.

All models have been implemented in PyTorch, architectures are described in appendix D. For the baselines PhyDNet, UV-CDN and PredRNN, we used the official source code provided by the authors. We evaluate on each scenario of **Filtered-CoPhy** on the counterfactual video prediction task. For the two counterfactual models (Ours and UV-CDN), we evaluate on the tasks as intended: we provide the observed sequence **AB** and the CF initial condition **C**, and forecast the sequence **D**. The non-CF baselines are required to predict the entire video from a single frame, in order to prevent them from leveraging shortcuts in a part of the video and bypass the need for physical reasoning.

We measure performance with time-averaged peak signal-to noise ratio (**PSNR**) that directly measures reconstruction quality. However, this metric is mainly dominated by error on the static background, which is not our main interest. We also introduce Localized PSNR (**L-PSNR**), which measures area error on the important regions near moving objects, computed on masked images. We compute the masks using classical background subtraction techniques.

**Comparison to the SOTA** — We compare our model against UV-CDN, PhyDNet and PredRNN in Table 1, consistently and significantly outperforming the baselines. The gap with UV-CDN is particularly interesting, as it confirms the choice of additional coefficients to model the dynamics of moving objects. PredRNN shows competitive performances, especially on `collisionCF`. However, our localized PSNR tends to indicates that the baseline does not reconstruct accurately the foreground, favoring the reconstruction of the background to the detriment of the dynamics of the scene. Fig. 6 visualizes the prediction on a single example, more can be found in appendix G. We also compare to trivial copying baselines in Table 2, namely Copy B, which assumes no intervention and outputs the **B** sequence, and Copy C, which assumes a stable tower. We evaluate these models in keypoints space measuring MSE on keypoints and coefficients averaged over time, as copying baselines are unbeatable in the regions of static background, making the PSNR metrics unusable.

We provide additional empirical results by comparing the models using Multi-object Tracking metrics and studies on the impact of the do-operations on PSNR in appendix E. We also compute an upper bound of our model using CoPhyNet baseline as described in Baradel et al. (2020).

**Performance on real-world data** — is reported in appendix F, showing experiments on 516 videos of real wooden blocks introduced in (Lerer et al., 2016).

**Impact of appearance coefficients** — are reported in Table 3, comparing to the baseline using a keypoint-only representation. The coefficients have a significant impact: even increasing the number of keypoints to compensate for the loss of information cannot overcome the advantage of disentangling positions and shapes, as done in our model. We provide a deeper analysis of the

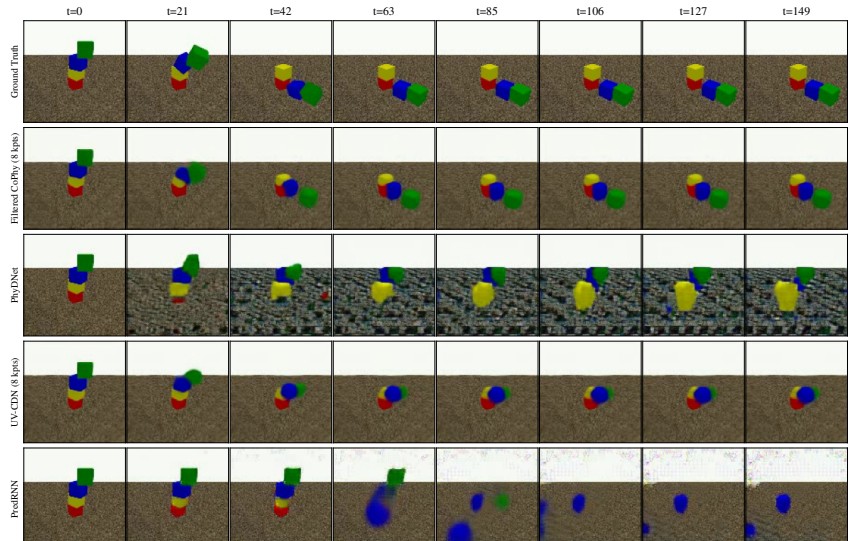

Figure 6: Visualization of the counterfactual video prediction quality, comparing our proposed model (Filtered CoPhy) with the two baselines, PhyDNet and UV-CDN, over different timestamps.

| State auto-encoder: | ✓ | ✗ |
|---|---|---|
| BT-CF | **9.58** | 11.10 |
| B-CF | **36.12** | 36.88 |
| C-CF | **5.14** | 16.16 |

| Filter bank: | Fixed | Learned |
|---|---|---|
| BT-CF | **34.40** | 32.04 |
| B-CF | **37.76** | 31.25 |
| C-CF | **34.09** | 33.88 |

Table 4: Impact of the dynamic CoDy encoder (✓) against the baseline operating in the keypoint + coefficient space (✗). We report MSE$\times 10^{-3}$ on prediction of keypoints + coefficients (4 pts).

Table 5: Learning the filter bank **H** from scratch has a mild negative effect on the reconstruction task. We report the PSNR on static reconstruction performance without the dynamic model.

de-rendering/rendering modules in appendix B, which includes visualizations of the navigation of the latent shape space in B.2.

**Learning filters** — does not have a positive impact on reconstruction performance compared to the choice of the handcrafted bank, as can be seen in table 5. We conjecture that the additional degrees of freedom are redundant with parts of the filter kernels in the refinement module $\mathcal{R}$: this corresponds to jointly learning a multi-channel representation $\{G_k^i\}_{k=1..K}^{i=1..C}$ for shapes as well as the mapping which geometrically distorts them into the target object shapes. Fixing the latent representation does not constrain the system, as the mapping $\mathcal{R}$ can adjust to it — see Fig. 5.

**Impact of the high-dimensional dynamic space** — We evaluate the impact of modeling object dynamics in high-dimensional space through the CoDy encoder in Table 4, comparing projection to 256 dimensions to the baseline reasoning directly in keypoint + coefficient space. The experiment confirms this choice of KKL-like encoder (Janny et al., 2021).

## 6 CONCLUSION

We introduced a new benchmark for counterfactual reasoning in physical processes requiring to perform video prediction, i.e. predicting raw pixel observations over a long horizon. The benchmark has been carefully designed and generated imposing constraints on identifiability and counterfactuality. We also propose a new method for counterfactual reasoning, which is based on a hybrid latent representation combining 2D keypoints and additional latent vectors encoding appearance and shape. We introduce an unsupervised learning algorithm for this representation, which does not require any supervision on confounders or other object properties and processes raw video. Counterfactual prediction of video frames remains a challenging task, and Filtered CoPhy still exhibits failures in maintaining rigid structures of objects over long prediction time-scales. We hope that our benchmark will inspire further breakthroughs in this domain.

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

# Appendix

## A  FURTHER DETAILS ON DATASET GENERATION

**Confounders** in our setup are masses, which we discretize in $\{1, 10\}$. For `BallsCF` and `CollisionCF`, we can also consider the continuous initial velocities of each object as confounders variables, since they have to be identified in **AB** to forecast **CD**.

We simulate all trajectories associated with the various possible combinations of masses from the same initial condition.

**Do-interventions**, however, depend on the task. For `BlocktowerCF` and `BallsCF`, do-interventions consist in (a) removing the top cube or a ball, or (b) shifting a cube/ball on the horizontal plane. In this case, for `BlocktowerCF`, we make sure that the cube does not move too far from the tower, in order to maintain contact. For `CollisionCF`, the do-interventions are restricted to shifting operations, since there are only two objects (a ball and a cylinder). It can consist of either a switch of the cylinder's orientation between vertical or horizontal, or a shift of the position of the moving object relative to the resting one in one of the three canonical directions $x, y$ and $z$.

### A.1  ENFORCING THE IDENTIFIABILTY CONSTRAINT

The identifiabilty and counterfactuality constraints described in section 3 are imposed numerically, i.e. we first sample and simulate trajectories with random parameters and then reject those that violate these constraints.

As stated in section 3, an identifiable experiment guarantees that there is no pair $(z, z')$ that gives the same trajectory **AB** but a different counterfactual outcome **CD**. Otherwise, there will be no way to choose between $z$ and $z'$ only from looking at **AB**, thus no way to correctly forecast the counterfactual experiment. By enforcing this constraint, we make sure that there exists at least a set $\{z, z', ...\}$ of confounders that give at the same time similar observed outcomes **AB** and similar counterfactual outcomes **CD**.

In practice, there exists a finite set of possible variables $z_i$, corresponding to every combination of masses for each object in the scene (masses take their value in $\{1, 10\}$). During generation, we submit each candidate experiment $(\mathbf{AB}, \mathbf{CD}, z)$ to a test ensuring that the candidate is identifiable. Let $\psi(X_0, z)$ be the function that gives the trajectory of a system with initial condition $X_0$ and confounders $z$. We simulate all possible trajectories $\psi(A, z_i)$ and $\psi(C, z_i)$ for every possible $z_i$. If there exists $z' \neq z$ such that the experiment is not identifiable, the candidate is rejected. This constraint requires to simulate the trajectory of each experiment several times by modifying physical properties of the objects.

Equalities in Definition 1 are relaxed by thresholding distances between trajectories. We reject a candidate experiment if there exists a $z'$ such that

$$\sum_{t=0}^{T} \|\psi(A, z) - \psi(A, z')\|_2 < \varepsilon \text{ and } \sum_{t=0}^{T} \|\psi(C, z) - \psi(C, z')\|_2 > \varepsilon. \tag{7}$$

The choice of the threshold value $\varepsilon$ is critical, in particular for the identifiability constraint:

- If the threshold is **too high**, all **AB** trajectories will be considered as equal, which results in acceptance of unidentifiable experiments.

- If the threshold is **too low**, all trajectories **CD** are considered equal. Again, this leads to mistakenly accepting unidentifiable experiments.

There exists an optimal value for $\varepsilon$, which allows to correctly reject unidentifiable experiences. To measure this optimal threshold, we generated a small instance of the `BlocktowerCF` dataset without constraining the experiments, i.e. trajectories can be unidentifiable and non-counterfactual. We then plot the percentage of rejected experiments in this unfiltered dataset against the threshold

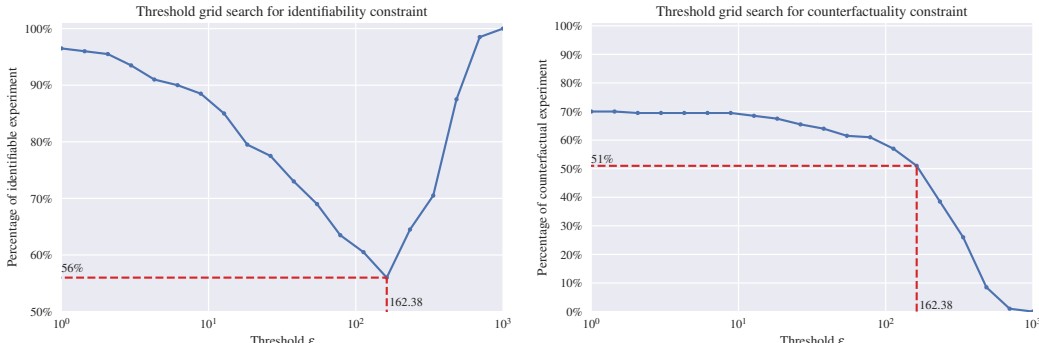

Figure 7: Experimental tuning of the threshold parameter. We generate an unconstrained subset of `BlocktowerCF` and plot the percentage of identifiable experiments as function of the threshold $\varepsilon$.

|  | without constraint | with constraint |
|---|---|---|
| Accuracy | 56% | **84%** |
| Corrected Acc. | 58% | **91%** |

(a)

| FPS | 5 | 15 | 25 | 35 | 45 |
|---|---|---|---|---|---|
| MSE ($\times 10^{-2}$) | 4.58 | 3.97 | **3.74** | 3.82 | 3.93 |

(b)

Table 6: (a) Sanity check of the identifiability constraint in `BlocktowerCF`, which results in better estimation of cube masses. The corrected accuracy only considers those cubes for which changes in masses are consequential for the trajectory $\mathbf{D}$. (b) MSE between ground truth 3D positions and predicted positions after 1 second, depending on the sampling rate of the trajectory.

value (Fig. 7, left). We chose the threshold $\varepsilon = 100$ which optimizes discrimination and rejects the highest number of "unidentifiable" trajectories.

To demonstrate importance of this, we train a recurrent graph network on `BlocktowerCF` to predict the cube masses from ground-truth state trajectories $\mathbf{AB}$, including pose and velocities, see Fig. 8. It predicts each cube's mass by solving a binary classification task. We train this model on both `BlocktowerCF` and an alternative version of the scenario generated without the identifiability constraint. The results are shown in Table 6a. We are not aiming 100% accuracy, and this problem remains difficult in a sense that the identifiability constraint ensures the identifiability of a set of confounder variables, while our sanity check tries to predict a unique $z$.

However, the addition of the identifiability constraint to the benchmark significantly improves the model's accuracy, which indicates that the property acts positively on the feasability of **Filtered-CoPhy**. The *corrected accuracy* metric focuses solely on the critical cubes, i.e. those cubes whose masses directly define the trajectory $\mathbf{CD}$.

## A.2 Enforcing the Counterfactuality constraint

Let $(\mathbf{AB}, \mathbf{CD}, z)$ be a candidate experiment, and $z^k$ be a combination of masses identical to $z$ except for the $k^{th}$ value. The counterfactuality constraint consists in checking that there exists at least one $k$ such that $\psi(C, z) \neq \psi(C, z^k)$. To do so, we simulate $\psi(C, z^k)$ for all $k$ and measure the difference with the candidate trajectory $\psi(C, z)$. Formally, we verify the existence of $k$ such that:

$$\sum_{t=0}^{T} \|\psi(C, z^k) - \psi(C, z)\|_2 < \varepsilon. \tag{8}$$

## A.3 Analyzing temporal resolution

We analyzed the choice of temporal frequency for the benchmark with another sanity check. We simulate a non-counterfactual dataset from `BlocktowerCF` where all cubes have equal masses. A

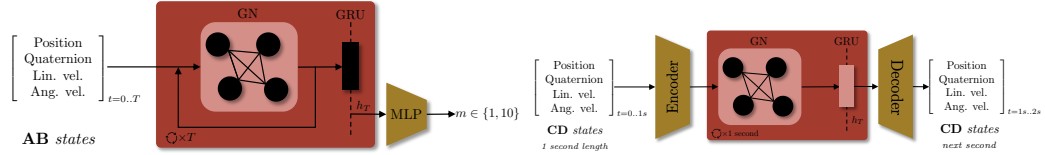

Figure 8: Impact of the choice of temporal resolution. Left: We check the identifiability constraint by training a model to predict the cube masses in `BlocktowerCF` from the observed trajectory **AB**. The model is a graph neural network followed by a gated recurrent unit. Right: We check the augmentation of the sampling rate by training an agent to forecast a 1 second-length trajectory from the states of the previous second.

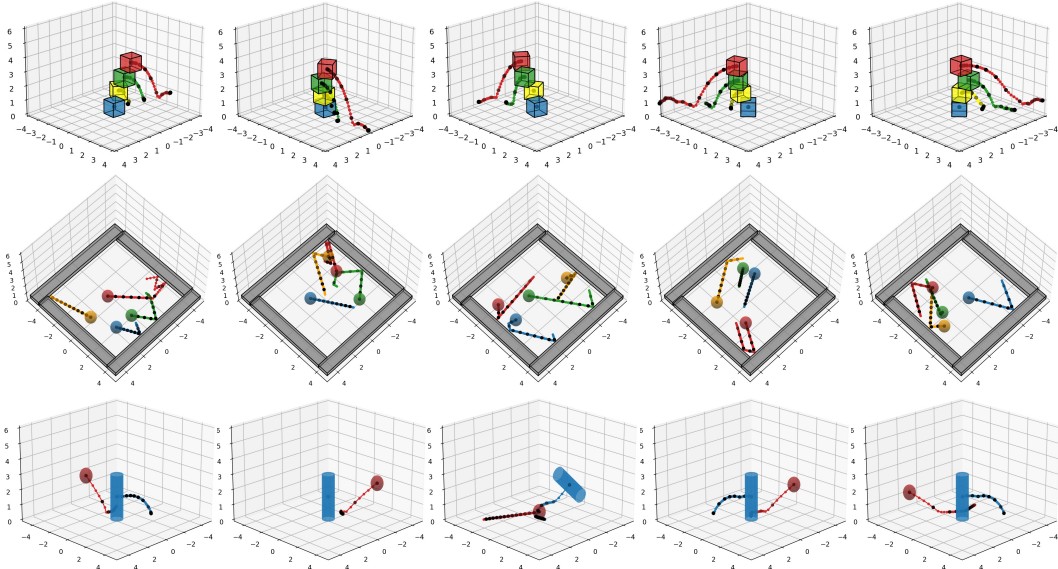

Figure 9: Visual examples of the impact of temporal resolution on dynamical information for each task in **Filtered-CoPhy**. Black dots are sampled at 6 FPS while red dots are sampled at 25 FPS.

recurrent graph network takes as input cube trajectories (poses and velocities) over a time interval of one second and predicts the rest of the trajectory over the following second. We vary the sampling frequency; for example, at 5 FPS, the model receives 5 measurements, and predicts the next 5 time steps, which correspond to a one second rollout in the future. Finally, we compare the error in 3D positions between the predictions and the GT on the last prediction. Results are shown in Table 6b. This check shows clearly that 25 FPS corresponds to the best trade-off between an accurate representation of the collision and the amount of training data. Fig. 9 shows a practical example of the effect of time resolution on dynamical information in a trajectory.

## A.4 SIMULATION DETAILS

We used Pybullet as physics engine to simulate **Filtered-CoPhy**. Each experiment is designed to respects the balance between good coverage of confounder combinations and counterfactuality and identifiability constraints described above. We generate the trajectories iteratively:

1. We sample a combination of masses and other physical characteristics of the given experiment, such as stability of the tower, or object motion in `CollisionCF`, or if the do-operation consists in removing an object. This allows us to maintain a balance of confounder configurations.

2. Then we search for an initial configuration **A**. For `BlocktowerCF`, we make sure that this configuration is unstable to ensure identifiability. Then we simulate the trajectory **B**.

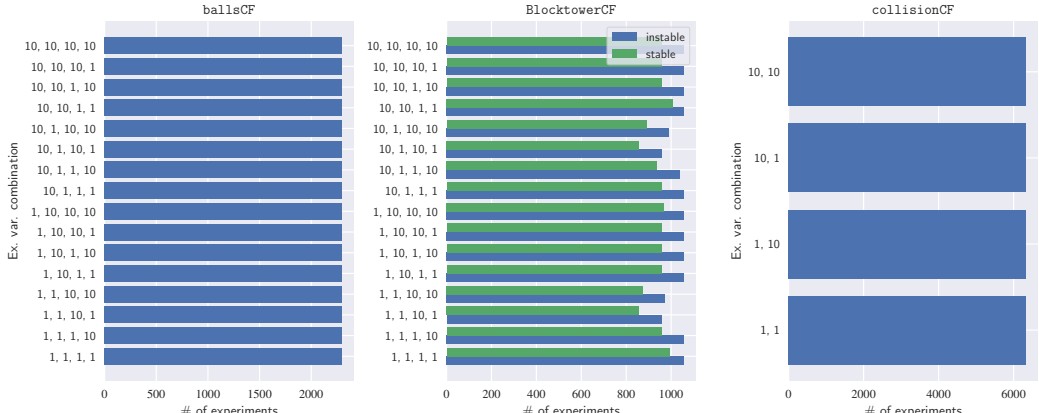

Figure 10: During the dataset generation process, we carefully balance combinations of masses, i.e. the confounders. For `BlocktowerCF`, we also guarantee that the proportion of stable **CD** towers is close to 50% for each confounder configuration.

3. We look for a valid do-operation such that identifiability and counterfactuality constraints are satisfied. If no valid do-operation is found after a fixed number of trials, we reject this experiment.

4. If a valid pair (**AB**, **CD**) is found, we add the sample to the dataset.

The trajectories were simulated with a sample time of $0.04$ seconds. The video resolution is $448 \times 448$ and represents 6 seconds for `BlocktowerCF` and `BallsCF`, and 3 seconds for `CollisionCF`. We invite interested readers to look at our code for more details, such as do-operation sampling, or intrinsic camera parameters. Fig. 10 shows the confounder distribution in the three tasks.

# B  PERFORMANCE EVALUATION OF THE DE-RENDERING MODULE

## B.1  IMAGE RECONSTRUCTION

We evaluate the reconstruction performance of the de-rendering module in the reconstruction task. Note that there is trade-off between the reconstruction performance and the dynamic forecasting accuracy: a higher number of keypoints may lead to better reconstruction, but can hurt prediction performance, as the dynamic model is more difficult to learn.

**Reconstruction error** – We first investigate the impact of the number of keypoints in Table 7 by measuring the Peak Signal to Noise Ratio (PSNR) between the target image and its reconstruction. We vary the number of keypoints among multiples of $N$, the maximum number of objects in the scene. Increasing the number

| # Keypoints | | N | 2N | 4N |
|---|---|---|---|---|
| BT-CF | PSNR | 34.40 | **35.41** | 34.92 |
| | MSE Grad | 27.24 | **21.39** | 23.99 |
| B-CF | PSNR | **37.76** | 37.06 | 36.98 |
| | MSE Grad | **3.47** | 3.77 | 3.95 |
| C-CF | PSNR | 32.00 | **35.41** | 34.42 |
| | MSE Grad | 32.00 | **12.57** | 17.09 |

Table 7: PSNR (dB) on the task of reconstructing the target from the source (both randomly sampled from **Filtered-CoPhy**), using 5 coefficients per keypoint. We vary the number of keypoints in our model. Here $N$ is the maximum number of the objects in the scene.

of keypoints increases reconstruction quality (PSNR) up to a certain point, but results in degradation in forecasting performance. Furthermore, doubling the number of keypoints only slightly improves reconstruction accuracy. This tends to indicate that our additional coefficients are already sufficient to model finer-grained visual details. Table 3 in the main paper measures the impact of the number of keypoints and the presence of the additional appearance coefficients on the full pipeline including the dynamic model. Table 8 illustrates the impact of the number of keypoints and the additional appearance coefficient on the reconstruction performance alone. As we can see, the addition of the coefficient consistently improves PSNR for low numbers of keypoints (over 2 dB for $N$ keypoints).

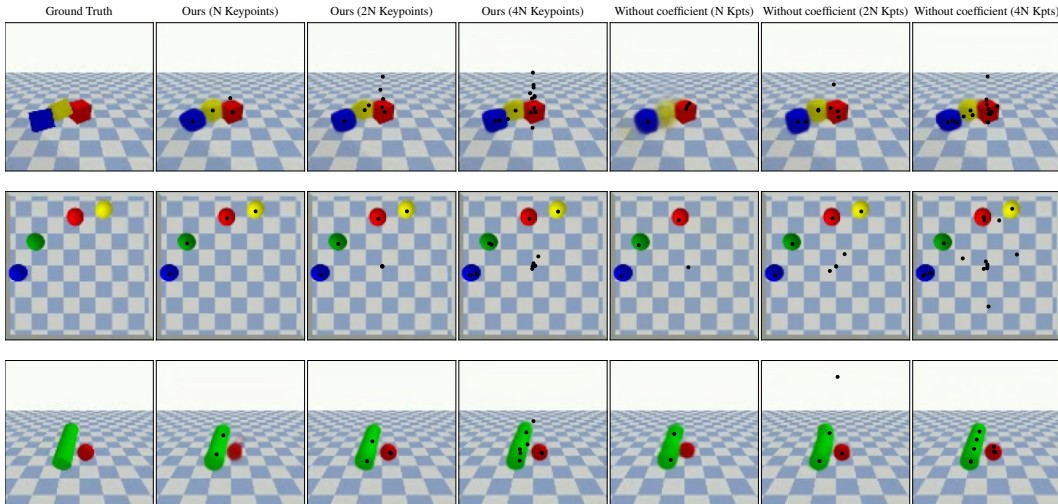

Figure 11: Reconstructions produced by the de-rendering module. Our model correctly marks each object in the scene and achieves satisfactory reconstruction.

The improvement is less visible for larger numbers of keypoints, since 3D visual details could be encoded via keypoints position, hence coefficients become less relevant. Visualizations are shown in Fig. 11.

| | | N | | 2N | | 4N | |
|---|---|---|---|---|---|---|---|
| Coefficients | | ✗ | ✓ | ✗ | ✓ | ✗ | ✓ |
| BT-CF | PSNR | 32.53 | **34.40** | 33.97 | **35.41** | 34.57 | **34.92** |
| | MSE Grad | 41.86 | **27.24** | 35.24 | **21.39** | 28.06 | **23.99** |
| B-CF | PSNR | 34.62 | **37.76** | 36.94 | **37.06** | **37.15** | 36.98 |
| | MSE Grad | 6.22 | **3.47** | 4.16 | **3.77** | **4.07** | 3.95 |
| C-CF | PSNR | 30.65 | **32.00** | 33.89 | **35.41** | **35.63** | 34.42 |
| | MSE Grad | 12.78 | **32.00** | 20.59 | **12.57** | **11.72** | 17.09 |

Table 8: Impact of the number of keypoints and the presence of additional appearance coefficient in the de-rendering module for pure image reconstruction (no dynamic model). We report PSNR (dB) and MSE on the image gradient. $N$ is the maximum number of the objects in the scene. The coefficients significantly improve the reconstruction on low number of keypoints. This table is related to table 3 in the main paper, which measures this impact on the full pipeline.

## B.2 NAVIGATING THE LATENT COEFFICIENT MANIFOLD

We evaluate the influence of the additional appearance coefficients on our de-rendering model by navigating its manifold. To do so, we sample a random pair $(X_{source}, X_{target})$ from an experiment in BlocktowerCF and compute the corresponding source features and target keypoints and coefficients. Then, we vary each component of the target keypoints and coefficients and observe the reconstructed image (fig. 12). We observed that the keypoints accurately control the position of the cube along both spatial axes. The rendering module does infer some hints on 3D shape information from the vertical position of the cube, exploiting a shortcut in learning. On the other hand, while not being supervised, the coefficients naturally learn to encode different orientations in space and distance from the camera. Interestingly, a form of disentanglement emerges. For example, coefficients n° 1 and 2 control rotation around the z-axis, and coefficient $n°4$ models rotation around the y-axis. The last coefficient represents both the size of the cube and its presence in the image.

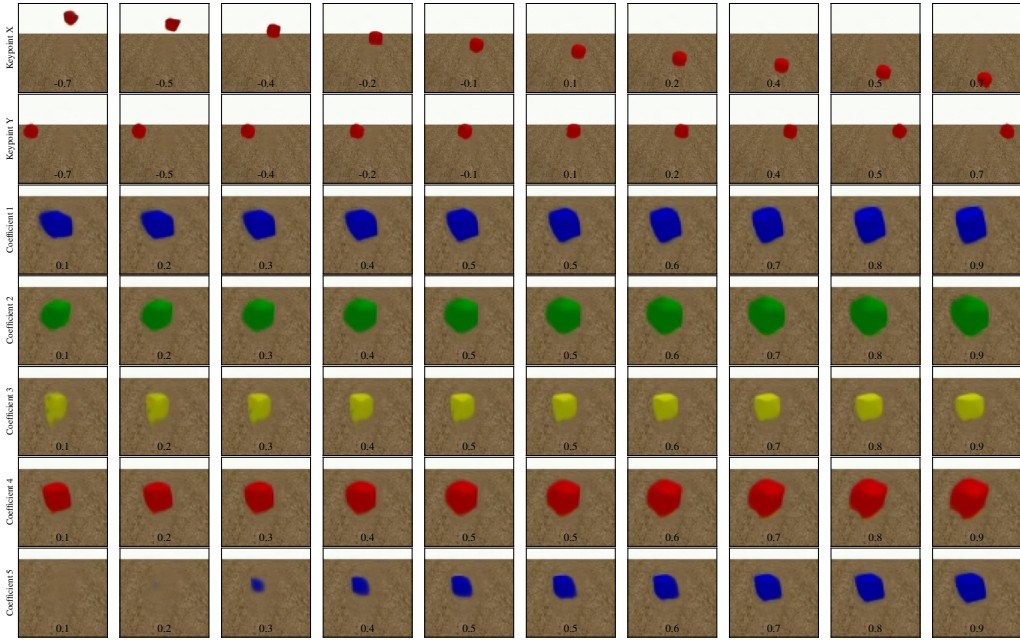

Figure 12: Navigating the manifold of the latent coefficient representation. Each line corresponds to variations of one keypoint coordinate or coefficient and shows the effect on a single cube.

## C   COMPARISON WITH THE TRANSPORTER BASELINE

### C.1   COMPARISON WITH OUR DE-RENDERING MODEL

As described in section 4.1, the Transporter (Kulkarni et al., 2019) is a keypoint detection model somewhat close to our de-rendering module. It leverages the transport equation to compute a reconstruction vector:

$$\hat{\Psi}_{\text{target}} = F_{\text{source}} \times (1 - K_{\text{source}}) \times (1 - K_{\text{target}}) + F_{\text{target}} \times K_{\text{target}}. \tag{9}$$

This equation allows to transmit information from the input by two means: the 2D position of the keypoints ($K_{\text{target}}$) and the dense visual features of the target ($F_{\text{target}}$). In comparison, our de-rendering solely relies on the keypoints from the target image and does not require a dense vector to be computed on the target to reconstruct the target. This makes the Transporter incomparable with our de-rendering module. We nevertheless compare the performances of the two models in Table 13, and provide visual examples in Fig. 13. Even though the two models are not comparable, as the Transporter uses additional information, our model still outperforms the Transporter for small numbers of keypoints. Interestingly, for higher numbers of keypoints Transporter tends to discover keypoints far from the object. We investigate this behavior in the following section, and show that this is actually a critical problem for learning causal reasoning on the discovered keypoints.

### C.2   ANALYSIS OF TRANSPORTER'S BEHAVIOR

The original version of the V-CDN model (Li et al., 2020) is based on Transporter (Kulkarni et al., 2019). We have already highlighted the fact that this model is not comparable with our task, as it requires not only the target keypoints $K_{\text{target}}$ but also a dense feature map $F_{\text{target}}$, whose dynamics can hardly be learned due to its high dimensionality. More precisely, the transport equation (Eq. 9) allows to pass information from the target by two means: the 2D position of the keypoints ($K_{\text{target}}$) and the dense feature map of the target ($F_{\text{target}}$). The number of keypoints therefore becomes a highly sensible parameter, as the transporter can decide to preferably transfer information through the target features rather than through the keypoint locations. When the number of keypoints is low, they act as a bottleneck, and the model has to carefully discover them to reconstruct the image.

|  | Ours | | | Transporter (not comparable) | | |
|---|---|---|---|---|---|---|
| # Keypoints | 4 | 8 | 16 | 4 | 8 | 16 |
| BT-CF | 34.40 | 35.41 | 34.92 | 34.10 | 34.88 | 39.20 |
| B-CF | 37.76 | 37.06 | 36.98 | 34.75 | 34.78 | 35.13 |
| C-CF | 35.41 | 34.42 | 35.98 | 32.66 | 33.39 | 34.47 |

Table 9: PSNR (dB) on the task of reconstructing target from the source (both randomly sampled from **Filtered-CoPhy**), using 5 coefficients per keypoint. We vary the number of keypoints in both our model and the Transporter. Note that Transporter uses target features to reconstruct the image, hence it is not comparable with our model.

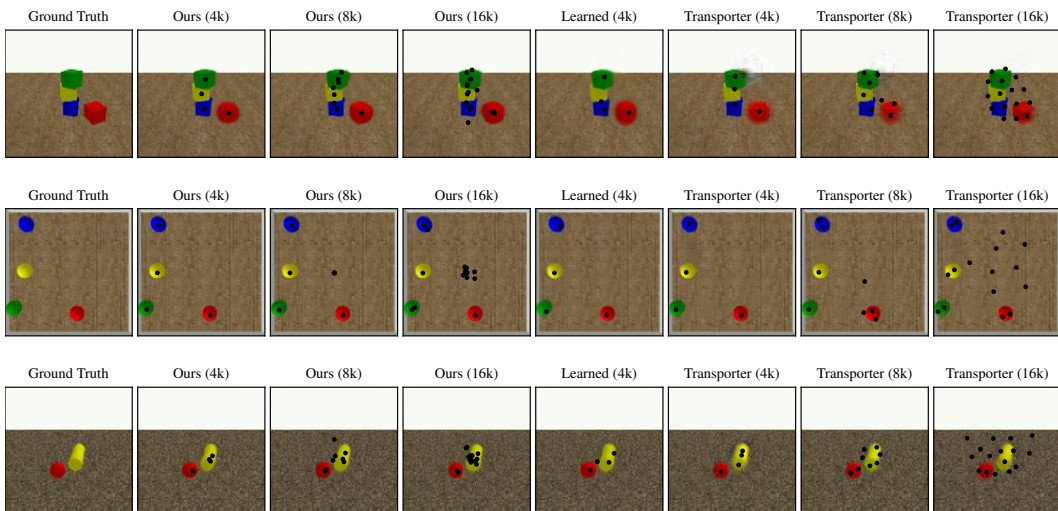

Figure 13: Example of reconstructed image by our-derendering module and the Transporter.

On the other hand, when we increase the number of keypoints, Transporter stops tracking objects in the scene and transfers visual information through the dense feature map, making the predicted keypoints unnecessary for image reconstruction, and therefore not representative of the dynamics.

To illustrate our hypothesis, we set up the following experiment. Starting from a trained Transporter model, we fixed the source image to be $X_0$ (the first frame from the trajectory) during the evaluation step. Then, we compute features and keypoints on the target frame $X_t$ regularly sampled in time. We reconstruct the target image using the transport equation, *but without updating the target keypoints*. Practically, this consists in computing $\hat{\Psi}_{\text{target}}$ with Eq. (9) substituting $K_{\text{target}}$ for $K_{\text{source}}$.

Results are shown in Fig. 14. There is no dynamic forecasting involved in this figure, and the Transporter we used was trained in a regular way, we only change the transport equation on evaluation time. Even though the keypoint positions have been fixed, the Transporter manages to reconstruct a significant part of the images, which indicates that a part of the dynamics has been encoded in the dense feature map.

In contrast, this issue does not arise from our de-rendering module, since our decoder solely relies on the target keypoints to reconstruct the image. Note that this is absolutely not contradictory with the claim in Li et al. (2020), since they do not evaluate V-CDN in pixel space. A rational choice of the number of keypoints leads to satisfactory performance, allowing V-CDN to accurately forecast the trajectory in keypoints space, and retrieve the hidden confounders on their dataset.

## C.3 TEMPORAL INCONSISTENCY ISSUES

Increasing the number of keypoints of the Transporter may lead to temporal inconsistency during the long-range reconstruction. For example, a keypoint that tracks the edge of a cube in the first

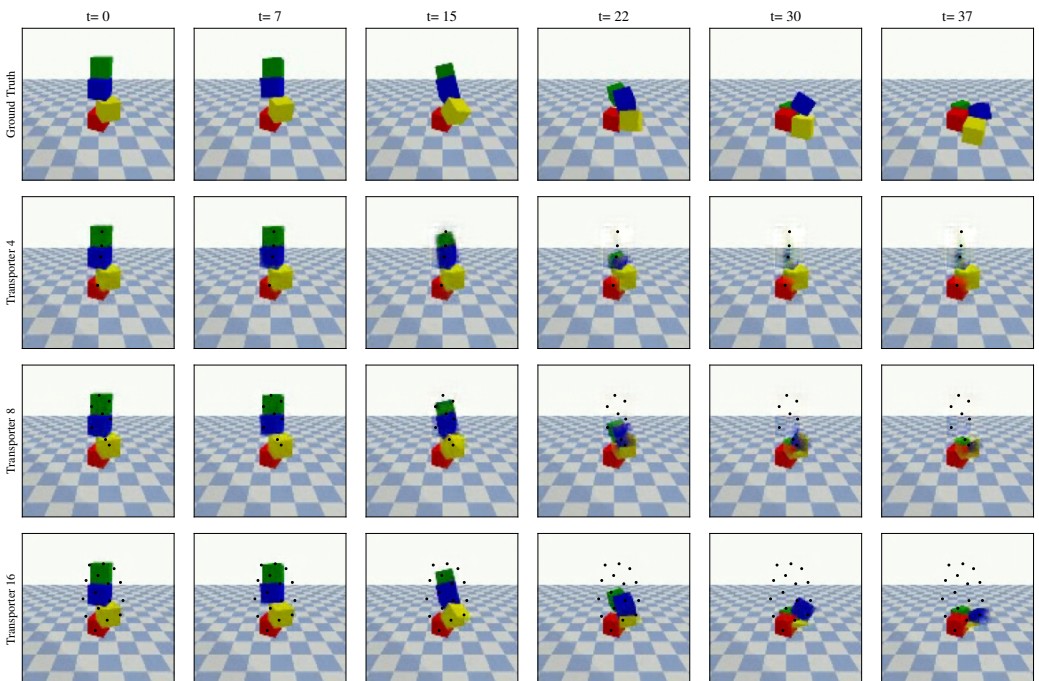

Figure 14: We evaluate the Transporter with a varying number of keypoints to reconstruct images regularly sampled in a trajectory while having the target keypoints fixed. Even if the keypoints are not moving, the Transporter still manages to reconstruct a significant part of the image, which indicates that the keypoints are not fully responsible for encoding the dynamics of the scene.

frame may target a face of this same cube in the future, since dynamics does not intervene in the keypoint discovery process.

Our de-rendering directly addresses this through the usage of additional appearance coefficients, which allows us to limit the number of keypoints to the number of objects in the scene, effectively alleviating the consistency issue. Fig. 15 illustrates this phenomenon by plotting the discovered keypoint locations forward in time, as well as the 2D location of the center of mass of each object. Note that the Transporter suffers from the temporal inconsistency issue with numbers of keypoints as low as 4 (green cube). In contrast, our model manages to solve the problem and accurately tracks the centers of mass, even though they were never supervised.

## D DETAILS OF MODEL ARCHITECTURES

### D.1 DE-RENDERING MODULE

We call a "block" a 2D convolutional layer followed by a 2D batch norm layer and ReLU activation. The exact architecture of each part of the encoder is described in Table 10a. The decoder hyper-parameters are described in Table 10b.

**Dense feature map estimator** $\mathcal{F}$ – We compute the feature vector from $\mathbf{X}_{\text{source}}$ by applying a convolutional network $\mathcal{F}$ on the output of the common CNN of the encoder. This produces the source feature vector $\mathbf{F}_{\text{source}}$ of shape (batch, 16, 28, 28).

**Keypoin detector** $\mathcal{K}$ – The convolutional network $\mathcal{K}$ outputs a set of 2D heatmaps of shape (batch, $K$, 28, 28), where $K$ is the desired number of keypoints. We apply a spatial softmax function on the two last dimensions, then we extract a pair of coordinates on each heatmap by looking for the location of the maximum, which gives us $\mathbf{K}_{\text{target}}$ of shape (batch, $K$, 2).

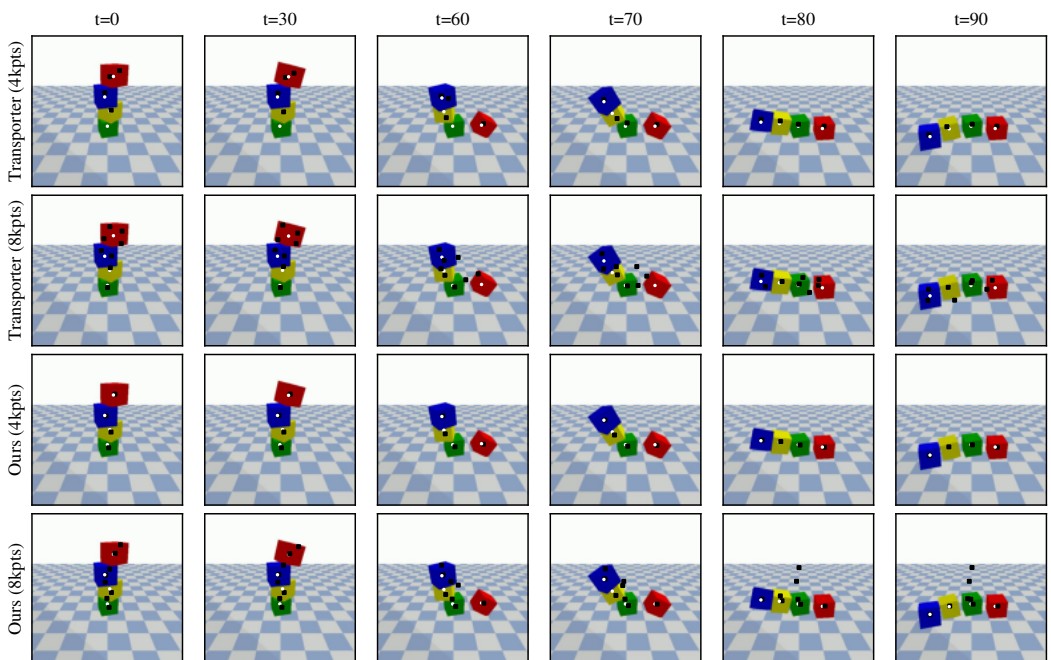

Figure 15: Temporal inconsistency in long-range reconstruction. We show the keypoints discovered on images taken from different time steps (black dots). We also compute the 2D location of the center of mass of each object in the scene (white dots). Our de-rendering module accurately tracks the centers of mass, which have never been supervised.

**Coefficient estimator** $\mathcal{C}$ – We obtain the coefficient by applying a third convolutional network $\mathcal{C}$ to the output of the common encoder CNN, which again results in a set of 2D vectors of shape $(\text{batch}, K, 28, 28)$. These vectors are flattened channel-wise and provide a tensor of shape $(\text{batch}, K, 28 \times 28))$ fed to an MLP (see Table 10a for the exact architecture) that estimates the coefficients $\mathbf{C}_{\text{target}}$ of shape $(\text{batch}, K, C + 1)$.

**Gaussian mapping** $\mathcal{G}$ – The keypoint vector $\mathbf{K}_{\text{target}}$ is mapped to a 2D vector through a Gaussian mapping process :

$$\mathcal{G}(\mathbf{k})(x, y) = \exp\left(-\frac{(x - \mathbf{k}_x)^2 + (y - \mathbf{k}_y)^2}{\sigma^2}\right), \tag{10}$$

where $\mathcal{G}(\mathbf{k}) \in \mathbb{R}^{28 \times 28}$ is the Gaussian mapping of the keypoint $\mathbf{k} = [\mathbf{k}_x \ \mathbf{k}_y]$. We deform these Gaussian mappings by applying convolutions with filters $\mathbf{H}_i$ controlled by the coefficients $\mathbf{c}_k^i$.

The filters from $\mathbf{H}$ are $5 \times 5$ kernels that elongate the Gaussian in a specific direction. Practically, we obtain the filter $\mathbf{H}_i$ by drawing a line crossing the center of the kernel and with a slope angle of $i\frac{\pi}{C}$ where $C$ is the number of coefficients. We then apply a 2D convolution :

$$\mathbf{G}_k^i = \mathbf{c}_k^{C+1} \mathbf{c}_k^i \left(\mathcal{G}(\mathbf{k}_k) * \mathbf{H}_i\right). \tag{11}$$

Note that we also compute a supplementary coefficient $\alpha_k^{C+1}$ used as a gate on the keypoints. By setting this coefficient to zero, the de-rendering module can de-activate a keypoint (which is redundant with deactivating the full set of coefficients for this keypoint).

**Refiner** $\mathcal{R}$ – To reconstruct the target image, we channel-wise stack feature vectors from the source with the constructed filters and feed them to the decoder CNN $\mathcal{R}$ (Table 10b).

We trained the de-rendering module on pairs of images $(\mathbf{X}_{\text{source}}, \mathbf{X}_{\text{target}})$ randomly sampled from sequences $\mathbf{D}$. For a given sequence $\mathbf{D}$, we take $T - 25$ first frames of the trajectory as a source (where $T$ is the number of frames in the video). The last 25 frames are used as a target. For evaluation, we take the $25^{th}$ frame as the source, and the $50^{th}$ frame as the target. We use Adam optimizer with a learning rate of $10^{-3}$, $\gamma_1 = 10^4$ and $\gamma_2 = 10^{-1}$ to minimize equation 4.

| CNN | | | | | |
|---|---|---|---|---|---|
| *Module* | *in ch.* | *out ch.* | *kernel* | *stride* | *pad.* |
| 1 | | 3 | 32 | 7 | 1 | 3 |
| 2 | | 32 | 32 | 3 | 1 | 1 |
| 3 | Block | 32 | 64 | 3 | 2 | 1 |
| 4 | | 64 | 64 | 3 | 1 | 1 |
| 5 | | 64 | 128 | 3 | 2 | 1 |

| $\mathcal{F}()$ | | | | | |
|---|---|---|---|---|---|
| 1 | Block | 128 | 16 | 3 | 1 | 1 |

| $\mathcal{K}()$ | | | | | |
|---|---|---|---|---|---|
| 1 | Block | 128 | 128 | 3 | 1 | 1 |
| 2 | Conv2d | 128 | $K$ | 3 | 1 | 1 |
| 3 | Softplus | | | | | |

| $\mathcal{C}()$ | | |
|---|---|---|
| 1 | Block | 128 | $K$ | 3 | 1 | 1 |
| 2 | Flatten | | |
| | *Module* | *in* | *out* |
| 3 | Linear+ReLU | 784 | 2048 |
| 4 | Linear+ReLU | 2048 | 1024 |
| 5 | Linear+ReLU | 1024 | 512 |
| 6 | Linear+ReLU | 512 | $C$ |
| 7 | Sigmoid | | |

(a) Encoder architecture

| $\mathcal{R}()$ | | | | | | |
|---|---|---|---|---|---|---|
| | | *in ch.* | *out ch.* | *kernel* | *stride* | *pad.* |
| 1 | | $16+K \times C$ | 128 | 3 | 1 | 1 |
| 2 | Block | 128 | 128 | 3 | 1 | 1 |
| 3 | | 128 | 64 | 3 | 1 | 1 |
| 4 | UpSamplingBilinear2d(2) | | | | | |
| 5 | Block | 64 | 64 | 3 | 1 | 1 |
| 6 | | 64 | 32 | 3 | 1 | 1 |
| 7 | UpSamplingBilinear2d(2) | | | | | |
| 8 | Block | 32 | 32 | 3 | 1 | 1 |
| 9 | | 32 | 32 | 7 | 1 | 1 |
| 10 | Conv2d | 32 | 3 | 1 | 1 | 1 |
| 11 | TanH | | | | | |

(b) Decoder architecture

Table 10: Architectural details of the de-rendering module.

## D.2 CoDy

We describe the architectural choices made in CoDy. Let

$$\mathbf{s}(t) = [\mathbf{k}_k \ \mathbf{c}_k^1 \ ... \ \mathbf{c}_k^{C+1} \ \dot{\mathbf{k}}_k \ \dot{\mathbf{c}}_k^1 \ ... \ \dot{\mathbf{c}}_k^{C+1}]_{k=1..K} \tag{12}$$

be the state representation of an image $\mathbf{X}_t$, composed of the $K$ keypoints 2D coordinates with their $C + 1$ coefficients. The time derivative of each component of the state is computed via an implicit Euler derivation scheme $\dot{\mathbf{k}}(t) = \mathbf{k}(t) - \mathbf{k}(t - 1)$. We use a subscript notation to distinguish the keypoints from $\mathbf{AB}$ and $\mathbf{CD}$.

**CF estimator** – The latent representation of the confounders is discovered from $\mathbf{s}^{\mathbf{AB}}$. The graph neural network from this module implements the message passing function $f()$ and the aggregation function $g()$ (see equation 5) by an MLP with 3 hidden layers of 64 neurons and ReLU activation unit. The resulting nodes embeddings $\mathbf{h}^{\mathbf{AB}}(t) = \mathcal{GN}(\mathbf{s}^{\mathbf{AB}}(t))$ belong to $\mathbb{R}^{128}$. We then apply a gated recurrent unit with 2 layers and a hidden vector of size 32 to each node in $\mathbf{h}^{\mathbf{AB}}(t)$ (sharing parameters between nodes). The last hidden vector is used as the latent representation of the confounders $u_k$.

**State encoder-decoder** – The state encoder is modeled as a $\mathcal{GN}$ where the message passing function and the aggregation function are MLPs with one hidden layer of 32 units. The encoded state $\boldsymbol{\sigma}^{\mathbf{CD}} = \mathcal{E}(s^{\mathbf{CD}})$ lies in $\mathbb{R}^{256}$. We perform dynamical prediction in this $\boldsymbol{\sigma}$ space, and then project back the forecasting in the keypoint space using a decoder. The decoder $\Delta(\boldsymbol{\sigma}(t))$ first applies a shared GRU with one layer and a hidden vector size of 256 to each keypoint $\boldsymbol{\sigma}_k(t)$, followed by a graph neural network with the same structure as the state encoder.

**Dynamic system** – Our dynamic system forecasts the future state $\hat{\boldsymbol{\sigma}}(t + 1)$ from the current estimation $\hat{\boldsymbol{\sigma}}(t)$ and the confounders $\mathbf{u} = [\mathbf{u}_1 \ ... \ \mathbf{u}_K]$. It first applies a graph neural network to the concatenated vector $[\hat{\boldsymbol{\sigma}}(t) \ \mathbf{u}]$. The message passing function $f$ and the aggregation function $g$ are MLPs with 3 hidden layers of 64 neurons and ReLU activation function. The resulting nodes embeddings $\mathcal{GN}((\sigma)(t))$ belong to $\mathbb{R}^{64}$ and are fed to a GRU sharing weights among each nodes with

2 layers and a hidden vector of size 64. This GRU updates the hidden vector $\mathbf{v^{CD}}(t) = [\mathbf{v}_1 \, ... \, \mathbf{v}_K]$, that is then used to compute a displacement vector with a linear transformation:

$$\hat{\boldsymbol{\sigma}}_k(t+1) = \boldsymbol{\sigma_k}(t) + \mathbf{W}\mathbf{v_k}(t) + \mathbf{b}. \qquad (13)$$

CoDy is trained using Adam to minimize equation 6 (learning rate $10^{-4}$, $\gamma_3 = 1$). We train each CoDy instance by providing it with fixed keypoints states $\mathbf{s^{AB}}(t)$ and the initial condition $\mathbf{s^{CD}}(t = 0)$ computed by our trained de-rendering module. CoDy first computes the latent confounder representation $\mathbf{u}_k$, and then projects the initial condition into the latent dynamic space $\boldsymbol{\sigma}(t = 0) = \mathcal{E}(\mathbf{s^{CD}}(t = 0))$. We apply the dynamic model multiple times in order to recursively forecast $T$ time steps from $\mathbf{CD}$. We then apply the decoder $\Delta(\hat{\boldsymbol{\sigma}}(t))$ to compute the trajectory in the keypoint space.

# E    ADDITIONAL QUANTITATIVE EVALUATION

## E.1    MULTI-OBJECT TRACKING METRICS

When the number of keypoints matches the number of objects in the scene, the keypoint detector naturally and in an unsupervised manner places keypoints near the center of mass of each object (see Figure 15). Leveraging this emerged property, we provide additional empirical demonstration of the accuracy of our model by computing classical Multi-Object Tracking (MOT) metrics. In particular, we computed the Multi-Object Tracking Precision (MOTP) and the Multi-Object Tracking (MOTA) as described in Bernardin & Stiefelhagen.

- **MOTA** requires to compute the number of missed objects (i.e. not tracked by a keypoints) and the number of false positives (i.e. keypoints that do not represent an actual object). MOTA takes values in $[-1, 1]$, where 1 represents perfect tracking:

$$\text{MOTA} = 1 - \frac{\sum_t m_t + f_t + s_t}{\sum_t g_t}, \qquad (14)$$

  where $m_t$ is the number of missed objects at time $t$, $f_t$ is the number of false positives at time $t$, $s_t$ is the number of swaps at time $t$, and $g_t$ is the number of objects at time $t$.

- **MOTP** is a measurement of the distance between the keypoints and the ground-truth centers of mass conditioned on the pairing process:

$$\text{MOTP} = \frac{\sum_{i,t} d_t^i}{\sum_t c_t}, \qquad (15)$$

  where $c_t$ is the number of accurately tracked objects at time $t$ and $d_t^i$ is the distance between the keypoint and the center of mass of the $i^{th}$ association {keypoints+center of mass}.

Note that these metrics are related: low MOTP indicates that the tracked objects are tracked precisely, and low MOTA indicates that many objects are missed. Thus, to be efficient, a model needs to achieve both low MOTP and high MOTA.

We also reported the performances of CoPhyNet (Baradel et al., 2020) that predicts counterfactual outcomes in euclidian space using the ground-truth 3D space. As it uses GT object positions during training, it is not comparable and should be considered as a soft upper bound of our method. We present our results in Table 11. This confirms the superiority of our method over UV-CDN in keypoint space. The upper bound CoPhyNet takes advantage of the non-ambiguous 3D representation modeled by the ground-truth state of the object of the scene.

Our method also outperforms CoPhyNet on the `ballsCF` task, probably due to two phenomena. First, `ballsCF` is the only 2D task of FilteredCoPhy. Thus, CoPhyNet does not have an advantage of using ground-truth 3D positions. Second, the state-encoder in CoDy projects the 2D position of each sphere in a space where the dynamics is easier to learn, probably by breaking the non-linearity of collisions.

|  |  | **Ours** | UV-CDN | *CoPhyNet (not comparable)* |
|---|---|---|---|---|
| BT-CF | MOTA ↑ | 0.46 | 0.16 | *0.44* |
|  | MOTP ↓ | 3.34 | 4.51 | *0.72* |
| B-CF | MOTA ↑ | -0.07 | -0.73 | *-0.16* |
|  | MOTP ↓ | 4.64 | 5.83 | *5.10* |
| C-CF | MOTA ↑ | -0.14 | -0.19 | *0.21* |
|  | MOTP ↓ | 6.35 | 6.35 | *4.37* |

Table 11: MOT metrics for different methods. While not comparable, we report the CoPhyNet performance as a soft upper bound. Our method and UV-CDN use one keypoint per object.
MOTA ↑: higher is better; MOTP ↓: lower is better;

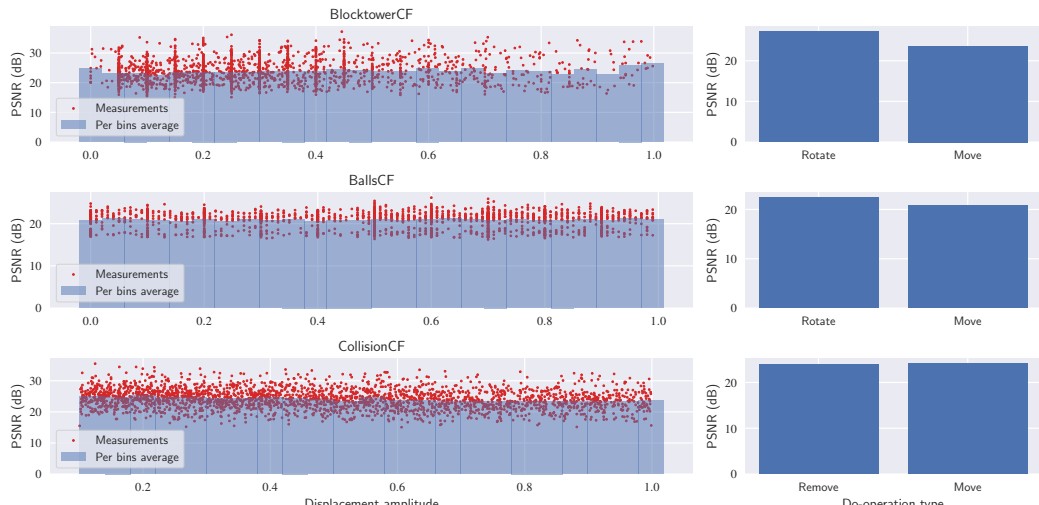

Figure 16: Effect of the do-operation on the quality of the forecasted video. (left) our method generalizes well to a wide range of "Move" operation amplitudes. (right) We observe a difference of 3dB in favor of the *Move* do-operation, which is unsurprising, as it is the least disturbing intervention

## E.2 IMPACT OF THE DO-OPERATIONS

We also measure the impact of the do-operation types on the video forecasting. Fig. 16 (left) is obtained by computing PSNR for each example of the training set and reporting the result on a 2D graph, depending on the amplitude of the displacement that characterizes the do-operation. We applied the same method to obtain Fig.16 (right) that focuses on the type of do-operation, that is moving, removing or rotating an object. These figures are computed using the 2N keypoints models.

Our method generalizes well across different do-operations, including both the type of the operation, and the amplitude. A key to this success it the careful design of the dataset (balanced with respect to the types of do-operations), and a reasonable representation (our set of keypoints and coefficients) able to detect and model each do-operation from images.

## F EXPERIMENTS ON REAL-WORLD DATA

Our contributions are focused on the discovery of causality in physics through counterfactual reasoning. We designed our model in order to solve the new benchmark and provided empirical evidence that our method is well suited for modeling rigid-body physics and counterfactual reasoning. The following section aims to demonstrate that our approach can also be extended to a real-world dataset. We provide qualitative results obtained on a derivative of `BlocktowerCF` using real cubes tower (Lerer et al., 2016).

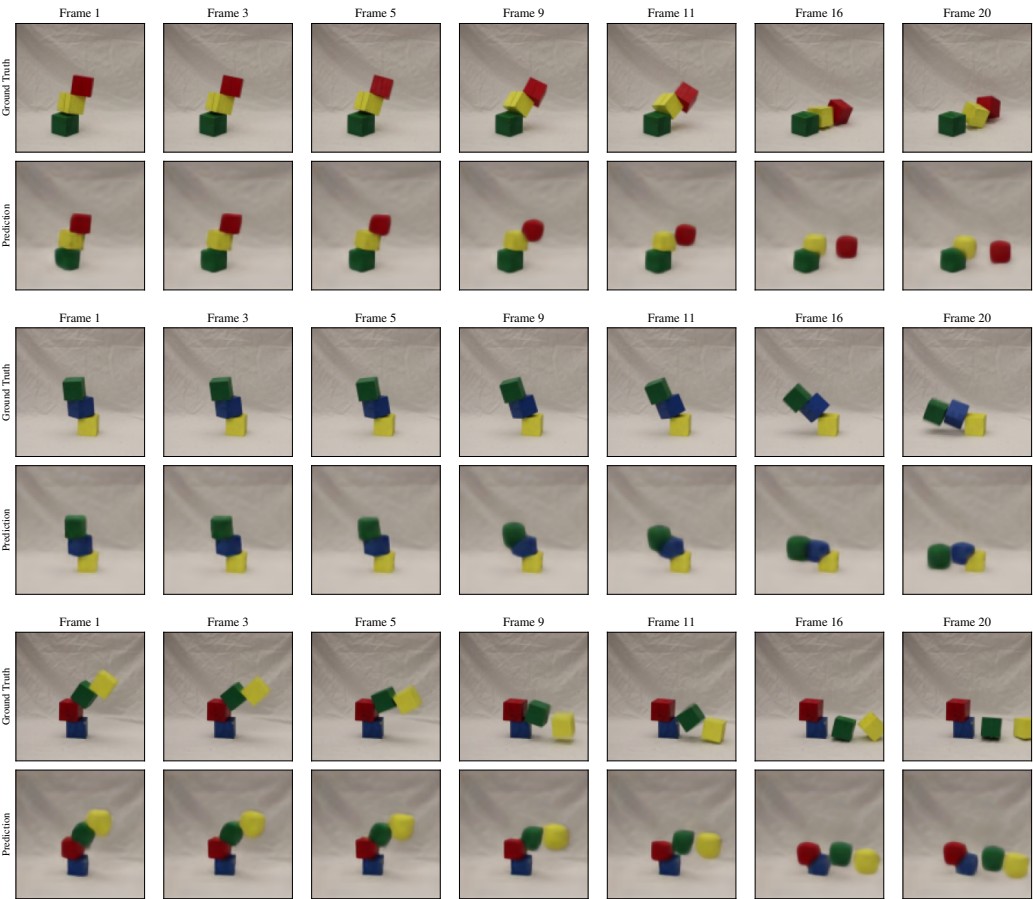

Figure 17: We evaluate our method on a real-world dataset Blocktower IRL. After fine-tuning, CoDy manages to accurately forecast future frames from real videos.

We refer to this dataset as *Blocktower IRL*. It is composed of 516 videos of wooden blocks stacked in a stable or unstable manner. The amount of cubes in a tower varies from 2 to 4. We aim to predict the dynamics of the tower in pixel space. This is highly related with our task `BlocktowerCF` (which inspired by the seminal work from (Lerer et al., 2016)) with three main differences: (1) the dataset shows real cube towers, (2) the problem is not counterfactual, i.e. every cube has the same mass and (3) the dataset contains only few videos.

To cope with the lack of data, we exploit our pre-trained models on `BlocktowerCF` and fine-tune on Blocktower IRL. The adaptation of the de-rendering module is straightforward: we choose the 4 keypoints-5 coefficients configuration and train the module for image reconstruction after loading the weights from previous training on our simulated task. CoDy, on the other hand, requires careful tuning to preserve the learned regularities from `BlocktowerCF` and prevent over-fitting. Since Blocktower IRL is not counterfactual, we de-activate the confounder estimator and set $u_k$ to vectors of ones. We also freeze the weights of the last layers of the MLPs in the dynamic model.

To the best of our knowledge, we are the first to use this dataset for video prediction. (Lerer et al., 2016) and Wu et al. (2017) leverage the video for stability prediction but actual trajectory forecasting was not the main objective. To quantitatively evaluate our method, we predict 20 frames in the future from a single image sampled in the trajectory. We measured an average PSNR of 26.27 dB, which is of the same order of magnitude compared to the results obtained in simulation. Figure 17 provides visual example of the output.

# G    QUALITATIVE EVALUATION: MORE VISUAL EXAMPLES

More qualitative results produced by our model on different tasks from our datasets are given below.

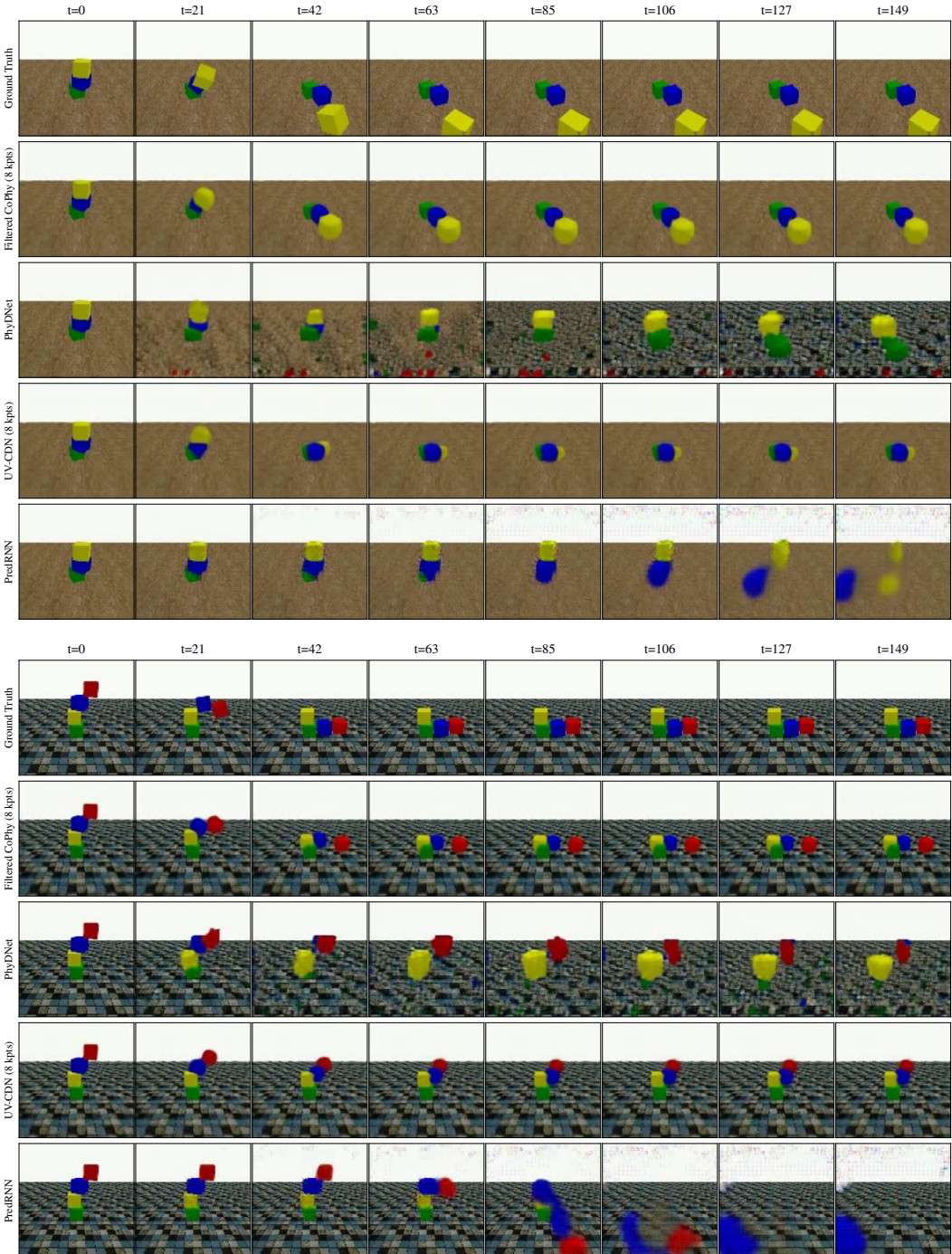

Figure 18: Qualitative performance on the `BlocktowerCF` (BT-CF) benchmark.

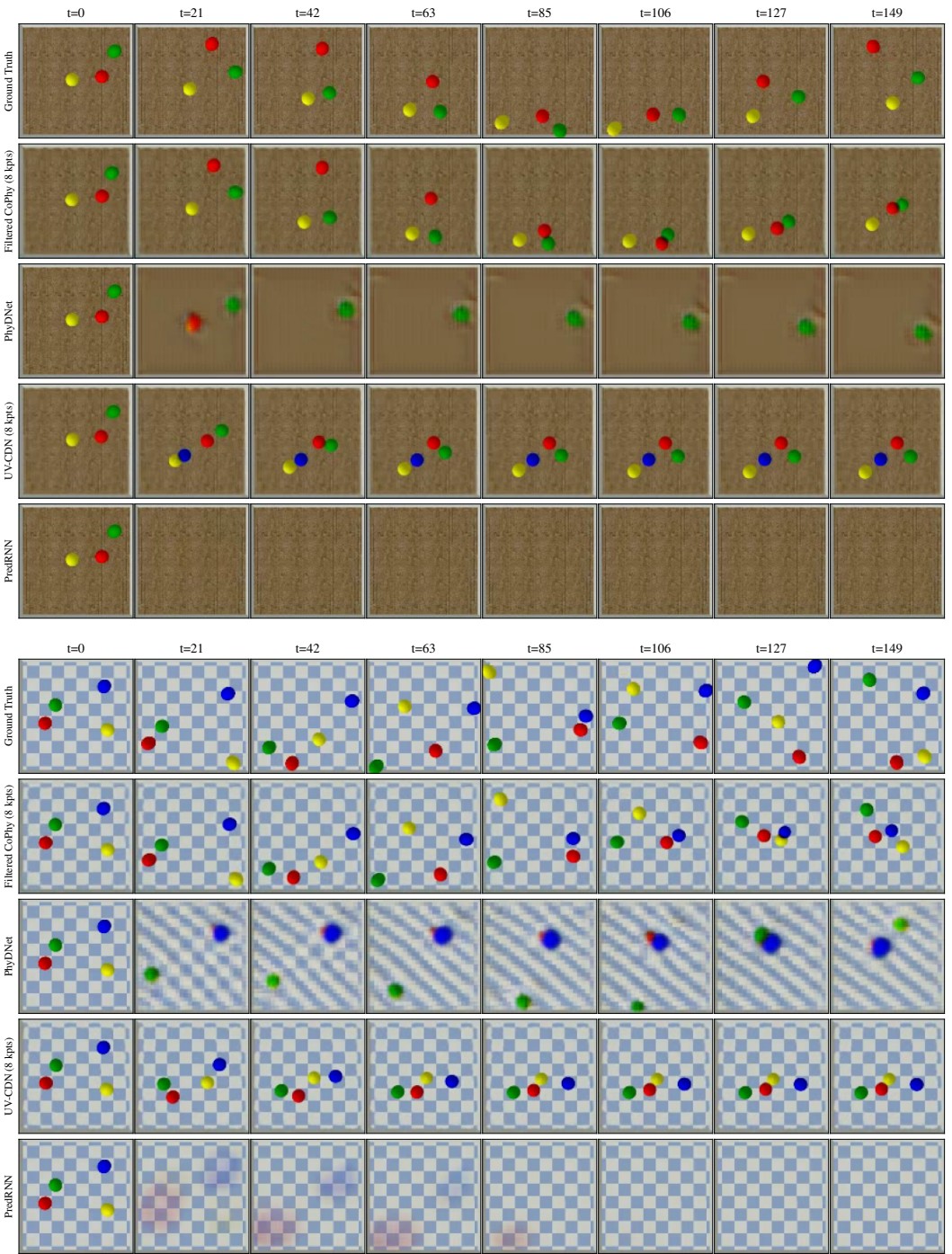

Figure 19: Qualitative performance on the `BallsCF` (B-CF) benchmark.

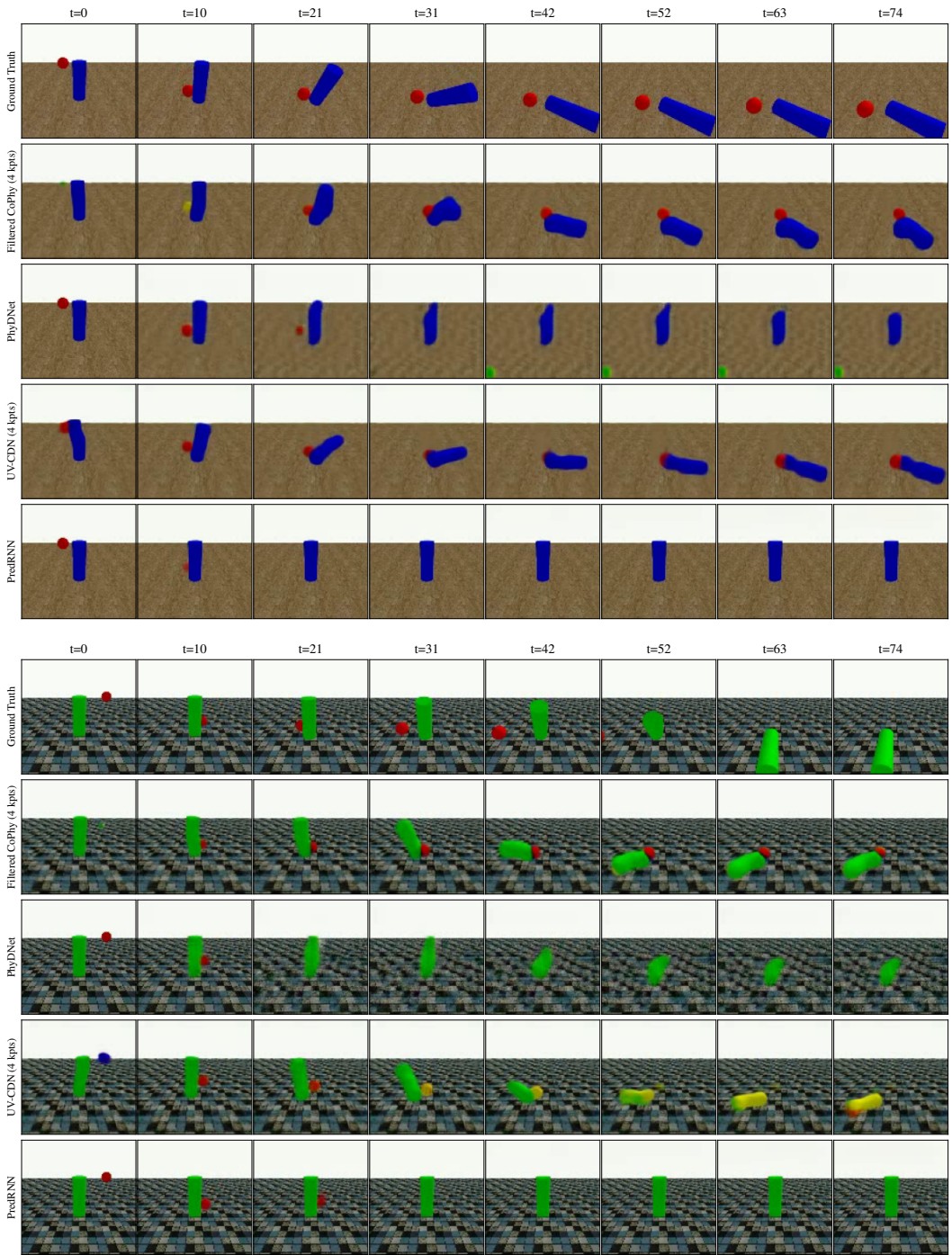

Figure 20: Qualitative performance on the `CollisionCF` (C-CF) benchmark.

