# OpenReview forum: "Filtered-CoPhy: Unsupervised Learning of Counterfactual Physics in Pixel Space"
_ICLR.cc/2022/Conference — ICLR 2022 Oral_

### Official Review · Reviewer_tY4H · 2021-11-01

**Correctness:** 4
**Technical Novelty And Significance:** 4
**Empirical Novelty And Significance:** 4
**Recommendation:** 10
**Confidence:** 3

**Main Review:**

Strengths
 - Introduces and justifies an important new benchmark task for video predictions involving inanimate objects
 - Introduces a sophisticated approach that is well-described and justified
 - Strong empirical evaluation that includes ablation studies and exploration of the learned representations
 - Thorough comparison with a previous approach that sheds light on reasons for the improvement
 - The counterfactual style of evaluation may be used in other domains of AI

Weaknesses
 - The paper would be even better if the diversity of the benchmark dataset could be improved.
 - Perhaps more could be said about the visible failures of the approach.  The network does not seem to succeed at learning the structure of rigid 3d bodies, as we see from the videos where the cubes visibly distort and lose their edges over successive frames.

Other comments
 - Many incomplete sentences in the supplement.  I would be happy to list them if this would be beneficial to the authors; I assume this was due to time limitations.

**Summary Of The Paper:**

The paper introduces a testing approach, dataset, and method for counterfactual video predictions, using 3d physics simulation videos.  Importantly, the approach predicts directly from pixel space rather than requiring spoon-fed keypoints.  It employs a counterfactual approach to establish a network's capacity to learn causal relations.  Separating the problem into one of parsing the inputs into keypoints and additional coefficients, inferring object attributes, and sequential prediction from an input frame + object attributes, it proposes an architecture which is based on combining modules for each of these learning tasks (but where keypoints are learned in an unsupervised way).  Besides this key architectural innovation, it adds an inductive bias by applying directional Gaussian filters to the keypoint maps.  The paper checks that the network actually works as intended by empirically examining the effect of changing object coefficients.  Three other ablation analyses, one that combines two of the modules (rather than separating them via stop-grad) and one that removes coefficients, and one comparing the handcrafted filter bank with learnable ones, add confidence to the approach.  In the supplement, the paper also analyzes in detail how the method compares to a previous method (Transporter).  It uses sensible benchmarks including previous methods and common-sense baselines where either the original or counterfactual input is used as a prediction, and does relatively well on this challenging problem, although the video on the website shows that there is still plenty of room for improvement.

**Summary Of The Review:**

This paper introduces an ambitious new baseline for counterfactual 3d physics predictions in pixel space.  It introduces a sophisticated method based on separate modules for image parsing, parameter inference, and prediction, that outperforms two previous approaches V-CDN and PhyDNet.

---

> ### Author Response · Authors · 2021-11-23
> **Answers to tY4H**
>
> Thank you for your valuable comments and appreciations. You clearly took the time to also read the quite long appendix, and we are delighted that you appreciate the extensive studies that we have conducted. Please find the answers of your questions below:
>
> **=== REMARK 1**: The paper would be even better if the diversity of the benchmark dataset could be improved.
>
> Generally speaking, we agree that more diversity would have made our benchmark even better. That said, there are some constraints on diversity, which derive from the properties we optimized: we generated a large-scale dataset with challenging regularities and clear definitions, integrating the constraints identifiability and counterfactuality, which we described in section 3 of the paper. We believe that these constraints are quite important, but they are also difficult to enforce, and adding even more diversity to the benchmark would have complicated this task. As said in the paper, in particular collisions are difficult to learn because their actions are both intense, brief, and highly non-linear, depending on the geometry of the objects in 3D space. Adding the additional problem of counterfactuality, which requires the unsupervised estimation and exploitation of confounders, makes the task even harder.
>
> We would also like to point out that, up to our knowledge, our dataset is the only counterfactual video dataset requiring predictions in pixel space. While CF reasoning is done in other work, it is classically restricted to classification problems.
>
> **=== REMARK 2**: Perhaps more could be said about the visible failures of the approach. The network does not seem to succeed at
> learning the structure of rigid 3d bodies, as we see from the videos where the cubes visibly distort and lose their edges over successive frames.
>
> You have correctly observed one of the failure cases of the method, in that objects have a slightly blurrier shape when they reach positions that are under-represented in the dataset, i.e. cubes that land very far from the original tower positions. In addition, we also observe some lack of rigidness when the prediction horizon is increased. This is particularly visible for the cylinder in the CollisionCF subset. Some animated examples are given on our project website: https://filteredcophy.github.io
>
> Finally, we also noted a bias toward lower frequencies in both the dynamical model and the derendering module. The predicted trajectories tend to be slightly smoother than the ground truth trajectories.

---

> > ### Comment · Reviewer_tY4H · 2021-11-25
> > **Additions to discussion?**
> >
> > I am quite impressed at the number of experiments you were willing and able to do in response to reviewer questions.  I do have two additional minor questions.
> >
> > 1. For section E.2, could you report the statistical tests used to conclude that "there is no significant difference of quality between do-operation types"?
> >
> > 2. Thanks for affirming my observations about the failures of the method.  I do understand that it may be difficult to perform given the amount of additional material you added to the paper in the revision, but could you include perhaps one or two sentences about these failure modes in the conclusion?  Something like "Counterfactual prediction of video frames remains a challenging task, and Filtered CoPhy still exhibits failures in maintaining rigid structures of objects over long prediction time-scales.  We hope that our benchmark will inspire further breakthroughs in this domain."

---

> > > ### Author Response · Authors · 2021-11-29
> > > **Discussions**
> > >
> > > Thank you for your quick response, and for your acknowledgement of our work to provide a convincing revision to our paper. Please find our answers to your questions below:
> > >
> > > **=== REMARK 1**: *For section E.2, could you report the statistical tests used to conclude that "there is no significant difference of quality between do-operation types"?*
> > >
> > > Thank you for your careful reading of our revised appendix. We actually do think that there might be differences in prediction performance according to intervention types, at least for certain tasks. We have made this clear in our initial author response, in particular to our answer to reviewer v4ms where we wrote “*We observe a difference of 3dB in favor of the “Move” do-operation, which is unsurprising, as it is the least disturbing intervention*“. However, we acknowledge a wrong statement in the caption of Figure 16 of the appendix, where we falsely claim that there is no significant difference.
> > > We also like to apologize for another small labeling error: the legends (labels) of the x-axis of the bar-graph of figure 16 should be inverted (“Remove” instead of “Rotate” and vice-versa).
> > > We will fix the caption and the figure on the camera-ready version of the paper, after acceptance.
> > > We also followed your request and performed a statistical test. To be precise, we computed the two-sided p-value for the null hypothesis that the PSNR distributions (averaged over time) of experiments done with (1) “Rotate/Remove” do-operations and, (2) “Move” do-operations are independant and have identical mean value but different standard deviation. The tests indicate differences for BlockTowerCF and BallsCF, but not for CollisionCF
> > >
> > >
> > >
> > > |         | BlocktowerCF |   BallsCF   | CollisionCF |
> > > |:-------:|:------------:|:-----------:|:-----------:|
> > > |         | Move/Remove  | Move/Remove | Rotate/Move |
> > > | p-value |   1.50e-94   |   2.12e-77  |     0.53    |
> > >
> > >
> > > Properly evaluating the impact of each do-operation on prediction quality is difficult, and quantitative measures are often misleading. We complemented them by qualitative measures: the visual demonstrations we put in the paper and on the project webpage allow the reader to assess the effect of each do-operation visually. We suggest that there are only small differences in quality between the different types of do-operations in, both, the predicted dynamics and the quality of the reconstruction of the objects themselves.
> > >
> > > **=== REMARK 2** : *Thanks for affirming my observations about the failures of the method. I do understand that it may be difficult to perform given the amount of additional material you added to the paper in the revision, but could you include perhaps one or two sentences about these failure modes in the conclusion? Something like "Counterfactual prediction of video frames remains a challenging task, and Filtered CoPhy still exhibits failures in maintaining rigid structures of objects over long prediction time-scales. We hope that our benchmark will inspire further breakthroughs in this domain."*
> > >
> > > We will integrate this into the camera ready version of the paper, after acceptance. We can’t do it immediately, since the submitted pdfs can’t be updated anymore since November 22th.

---

> > > > ### Comment · Reviewer_tY4H · 2021-11-30
> > > > **Thanks**
> > > >
> > > > I appreciate your work looking into the significance of the results in E.2 and your willingness to incorporate my suggestion in the conclusions.

---

### Official Review · Reviewer_sqvy · 2021-11-02

**Correctness:** 4
**Technical Novelty And Significance:** 3
**Empirical Novelty And Significance:** 3
**Recommendation:** 8
**Confidence:** 4

**Main Review:**

Strengths:
1) Filtered-CoPhy seems to be the next intuitive step from CoPhy from a modeling perspective. It is incremental work and a crucial one to get counterfactual predictions (for physics-based simulations) working in an unsupervised fashion.

2) The new benchmark proposal considering Identifiability and Counterfactuality constraints is systematic and alleviates the issues in the CoPhy benchmark.

3) I acknowledge the code released during the review period by the author(s). I have had a chance to briefly look over it and hope that the authors document the code (with a README and instructions on how to run the code) if the paper is accepted.


Weakness:

1) Keypoint based encoder/decoder seems to be a good choice. Have the author(s) tried/given a thought how other object-centric representations such as slot-attention would affect the counterfactual performance?

2) It would make the paper stronger if the CoPhy like baseline is reported i.e one using ground truth object positions. This would give at least an upper bound for the proposed benchmark.

3) It would be beneficial to add a quantitative metric based on the predicted and ground truth location of the center of mass (for both Table 1 and Table 3).

4) Also, since the keypoint points can be tracked, it would add more credibility if tracking metrics such as MOT are added as well. Here the author(s) can assume N=number of objects in the scene (otherwise this metric cannot be computed).

5) During the training stage, can the sequences AB be sent to the CoDy (i.e dynamics model) as well? Since $u_k$ should be the same for AB and CD, this acts as a consistency check.

**Summary Of The Paper:**

This work extends CoPhy, in which the author(s) address the problem of predicting counterfactual outcomes of physics-based tasks from pixel space (CoPhy used ground truth object positions). To do this, they learn a keypoint representation of the scene and use them to extract the confounders (such as velocities of objects, masses, etc). The author(s) also propose a new benchmark (built upon CoPhy's benchmark) for counterfactual prediction (after intervening the initial set of objects in the scene) which satisfies the Identifiability and the Counterfactuality constraints of causality.

**Summary Of The Review:**

Overall I feel that the paper is a strong contribution towards visual counterfactual reasoning (making the framework unsupervised as opposed to CoPhy) and would strongly urge the author(s) to add the experiments mentioned in the Weaknesses section. Based on the novelty of the paper, I am voting for a weak acceptance of the paper.

=======
Post-rebuttal decision:
After reading the comments of the authors as well as reviews of the other reviewers, I'm very much satisfied with the additional experiments added in the paper and I believe this paper would be a good contribution towards unsupervised visual counterfactual prediction. Hence, I've increased my score to 8 (Accept).

---

> ### Author Response · Authors · 2021-11-23
> **Answers to sqvy**
>
> Thank you for your valuable comments and appreciations. Please find the answers of your questions below:
>
> **=== REMARK 1**: Have the author(s) tried/given a thought how other object-centric representations such as slot-attention would affect the counterfactual performance?
>
> Yes, this is an interesting point, since keypoint detection is very close to slot-attention mechanisms. Using slot-attention is future work that we are interested in pursuing. We would like to point out that slot-attention has been proposed for static images, and we are not aware of published work which has extended it to RGB input video.
>
> **=== REMARK 2**: It would make the paper stronger if the CoPhy like baseline is reported i.e one using ground truth object positions. This would give at least an upper bound for the proposed benchmark.
>
> We added a comparison with the CoPhy like baseline as an upper bound. However, while our keypoint detector indeed does discover points close to the center of mass of each object, as the method is not unsupervised, there is no explicit enforcement of this rule: the de-rendering module could perfectly learn a keypoint representation far from the natural choice. However, the CoPhyNet error published in the original paper is measured as Mean squared error between centers of mass, whereas CoDy error is measured as MSE between keypoints, so the results would not have been comparable.
> Thus, as you suggested in another comment, MOT metrics should be particularly relevant for this task, as they evaluate how well an object is tracked, and this allows us to propose comparisons to CoPhyNet. Our results are provided in the revised appendix E.1, Table 11. We evaluate CoPhyNet, the model introduced in Baradel et al., on the FilteredCoPhy benchmark. We trained CoPhyNet by providing it with ground truth 3D positions of the objects in A, B and C, and minimizing the MSE between the output of the model and the positions in D. To evaluate CoPhyNet in the same domain as CoDy, we project the prediction on the camera plane (using the parameters of the camera), and then compute the MOT metrics between the projected predictions and the projected ground truth CoMs.
>
> **=== REMARK 3**: It would be beneficial to add a quantitative metric based on the predicted and ground truth location of the center of mass (for both Table 1 and Table 3).
>
> This is indeed something that deserves more experiments. Figure 15 in the appendix suggests that discovered keypoints are close to the ground truth center of mass of the objects. Interestingly, and as said above, this constraint is (a) not enforced in any manner in our model and (b) absolutely not necessary for accurate image reconstruction. Following your advice, we completed Table 1 of the paper by measuring the error related to the CoM of the objects. You will find the results in the revised appendix E.1, where we compare MOTA and MOTP metrics between our method, UV-CDN and the soft upper bound CoPhyNet as mentioned above.  Using the same metrics for table 3 is not straightforward: since there are more keypoints than objects we would have to modify metrics definition to allow one-to-many association between keypoints and CoM. Thus, we decided to report measurements solely for configurations where the number of keypoints matches the number of objects.
>
> **=== REMARK 4**: Also, since the keypoint points can be tracked, it would add more credibility if tracking metrics such as MOT are added as well.
>
> We used MOTP and MOTA metrics to answer your previous question, and reported the results in the revised appendix.
>
> **=== REMARK 5**: During the training stage, can the sequences AB be sent to the CoDy (i.e dynamics model) as well? Since uk should be the same for AB and CD, this acts as a consistency check.
>
> Sending A to CoDy in order to predict B is absolutely feasible. It can be related to a data augmentation technique to improve the dynamical model in CoDy. Nonetheless, this method is not necessary for FilteredCoPhy, which contains enough experiments to represent a significant part of the data distribution. To validate this claim, we train CoDy using this trick on blocktowerCF. We did not observe a significant improvement concerning the MSE on keypoints (from 9.58e-3 without the trick of AB to 9.25e-3 with the trick).
>
> **=== REMARK 6**: I acknowledge the code released during the review period by the author(s). I have had a chance to briefly look over it and hope that the authors document the code (with a README and instructions on how to run the code) if the paper is accepted.
>
> We are sorry for not having had the README ready before. We now updated the repository with explanations and instructions for dataset generation, training and evaluation of our methods. We also worked on the code to make it easier to use, add keypoints generation and evaluation scripts (including MOT metrics implementation) and pre-trained models.

---

### Official Review · Reviewer_v4mS · 2021-11-03

**Correctness:** 4
**Technical Novelty And Significance:** 3
**Empirical Novelty And Significance:** 4
**Recommendation:** 8
**Confidence:** 5

**Main Review:**

Strengths:
- The paper is well-written.
- The proposed encoding-decoding framework and the corresponding two-stage learning process are reasonable.
- the newly designed representation (i.e, high-dimensional features + 2D keypoints + coefficients) is shown to facilitate unsupervised learning from pixel space.
- The paper provides good empirical results and extensive ablation studies for the effectiveness of each model component.

Weaknesses:
- The authors may give more empirical analyses about the impact of using different types of interventions.
- The proposed method is only compared with two existing approaches, including PhyDNet and a modified version of V-CDN. It would be better if the authors could include more existing approaches into the experimental comparison, for example, the keypoint-based model [Minderer et al., 2019], the VAE-based stochastic model [Ref1], and the ConvLSTM-based deterministic model [Ref2].
- All experiments are conducted in a simulated environment based on CoPhy, which is somewhat insufficient compared with previous video prediction literature (though I understand the paper studies a new problem). Since the proposed method does not require the supervision of confounders, I wonder if it can be generalized to real-world datasets such as Human3.6M, KTH, or BAIR robot pushing.

[Ref1] Stochastic Video Generation with a Learned Prior. ICML 2018.

[Ref2] PredRNN: Recurrent Neural Networks for Predictive Learning Using Spatiotemporal LSTMs. NIPS 2017.



**Summary Of The Paper:**

This paper studies an interesting problem of counterfactual video prediction, which aims to predict the future frame (D) based on the initial frame (C) and an observed video sequence (AB). AB can be seen as a demonstration that is driven by the same confounders, i.e., the physics parameters such as mass and initial velocities.

The paper follows the previous models from CoPhyNet for dynamics modeling and confounder estimation, and has two improvements over the work of CoPhy, in my point of view:
- First, it improves the CoPhy benchmark.
- Second, it presents a new model based on the representations of high-dimensional features, 2D keypoints, and corresponding coefficients. The new form of representation allows the model to be trained in an unsupervised manner only with the supervision of RGB images, as opposed to the training procedure with the supervision of object positions in CoPhyNet.



**Summary Of The Review:**

This paper explores an interesting problem of learning counterfactual physics from pixels. It presents a solution by extending a previous work, CoPhyNet, with new forms of unsupervised video representations.

One of my concerns is that it does not provide sufficient empirical comparisons with existing video prediction models other than PhyDNet and the modified V-CDN. Furthermore, I would increase my score if the effectiveness of the method can be validated in a real-world dataset.

---

> ### Author Response · Authors · 2021-11-23
> **Answers to v4ms**
>
> Thank you for your valuable comments and appreciations. Please find the answers of your questions below:
>
> **=== REMARK 1**: *The authors may give more empirical analyses about the impact of using different types of interventions.*
>
> We provided additional empirical analyses in appendix E.2. We conducted two experiments: (1) we measured the overall PSNR depending on the range of the do-operation. Our model performs almost uniformly on the displacement distribution, which shows robustness to the do-operation. (2) we measure the PSNR for each do-operation type, ie. ‘Remove’ and ‘Move’. We observe a difference of 3dB in favor of the “Move” do-operation, which is unsurprising, as it is the less disturbing intervention.
>
> **=== REMARK 2**: *The proposed method is only compared with two existing approaches, including PhyDNet and a modified version of V-CDN. It would be better if the authors could include more existing approaches into the experimental comparison, for example, the keypoint-based model, the VAE-based stochastic model, and the ConvLSTM-based deterministic model.*
>
> We added additional baselines, but we would like to first explain general troubles in identifying truly comparable baselines to assess the reliability of our method. This is due to the nature of the task: our benchmark requires to predict an entire video from a single image, and the knowledge of another video. To the best of our knowledge, VCDN is the most suited comparison baseline, but still needs access to ground truth data, (this was extensively studied and discussed in the appendix of the paper). Video generation methods appear at first glance a reasonable choice for comparison methods. Nonetheless, most of them leverage a known part of the video to forecast the rest of it. This is actually critical for our benchmark : the knowledge of a small (dynamical) part of CD may alleviate the need for (a) physical reasoning and (b) counterfactual reasoning. They can be avoided simply by pursuing the initial movement from the known part of the video, e.g. a stable block tower will be trivial to forecast knowing that the tower is stable during the first second.
>
> Nonetheless, we agree that adding methods from the video generation literature will be a valuable addition to our paper. We trained the VAE-based stochastic model (SVG-LP) and the ConvLSTM-based model (PredRNN) on each task of our dataset, doing our best to adapt them to the counterfactual task at hand, for which they were not designed.
>
> Despite our effort, the stochastic baseline failed to solve the FilteredCoPhy task and provided unusable results. We explored different configurations, hyperparameters and seeds. We decided not to add SVG-LP to the baselines in the paper, as the results were too bad to be shown. This method does not seem to be suited for solving this kind of tasks. PredRNN indeed appears to be a challenging baseline. However, while demonstrating excellent capacity to reconstruct background, it appears that the method fails to accurately track the objects forward in time This is one of the strengths of our methods: the object centric representation is made explicit by the use of keypoints. We reported our experiments in table 1 (revised main paper) and provided visualizations in figures 6 (revised main paper) and 18, 19, 20 (revised appendix).
>
> **=== REMARK 3**:  *All experiments are conducted in a simulated environment based on CoPhy, which is somewhat insufficient compared with previous video prediction literature (though I understand the paper studies a new problem). Since the proposed method does not require the supervision of confounders, I wonder if it can be generalized to real-world datasets such as Human3.6M, KTH, or BAIR robot pushing.*
>
> This is indeed an interesting study to conduct, although real datasets do not provide the leverage we have to target real counterfactual tasks with a control over identifiability, counterfactuality, etc. (see discussion in section 3). Given the short time available during the rebuttal period, we focused on one additional real-world dataset using real physical cubes (Lerer et al, ICML'16). More details are given in the revised appendix of the paper (appendix F).
>
> To the best of our knowledge, we are the first to use this dataset for video prediction, as (Lerer et al.,2016) and (Wu et al. 2017) use it for stability prediction, while actual trajectory forecasting was not their objective.
>
> The dataset consists of 516 videos of towers in stable or unstable configuration. The difficulty with this dataset lies in the low quantity of video available for training. Thus, we leverage the models trained on Blocktower from our own FilteredCoPhy benchmark and fine-tune the models on the videos from this dataset. We provide empirical results in Appendix F of the revised version of our paper. We observed satisfactory forecasting in pixel space, indicating that our method is transferable to other tasks in real life scenarios in a short time.

---

> > ### Comment · Reviewer_v4mS · 2021-11-24
> > **Waiting for the comparisons on the real-world dataset.**
> >
> > I sincerely appreciate the detailed explanations and the new results on the Block Tower dataset. The qualitative results look good. Then, could you please provide some comparisons with other existing work on this new dataset (at least the compared models used on FilteredCoPhy)?
> >
> > With the positive results, I would like to consider increasing my rating.

---

> > > ### Author Response · Authors · 2021-11-29
> > > **Comparisons on the real-world dataset**
> > >
> > > Thank you for your quick response, please find our answer to your question below :
> > >
> > > **=== REMARK**: *I sincerely appreciate the detailed explanations and the new results on the Block Tower dataset. The qualitative results look good. Then, could you please provide some comparisons with other existing work on this new dataset (at least the compared models used on FilteredCoPhy)?*
> > >
> > > Thank you for acknowledging our work to follow the reviewer's advice and improve our submission. As we said in the revised appendix F, there is little work around the Block Tower dataset, and to the best of our knowledge, we are the first to target video forecasting on this task. There are actually two published papers that perform experiments on this dataset. Lerer & al (2017) introduced the dataset as a validation for their stability prediction model. Forward prediction is used as a proxy task to encourage physical reasoning. Yet, predictions are not evaluated in a quantitative manner and performed over segmentation images with simulation data. Wu et al. (2017) also performs stability prediction, but they use an explicit physics engine to predict stability from a single frame.
> > >
> > > However, yes, comparing with the baselines methods we proposed for the FilteredCoPhy benchmark is feasible. We train PhyDNet, UV-CDN (4 keypoints) and PredRNN on this task. We followed the same procedure as we did with our methods: we tried to finetune each model pre-trained in simulation by freezing different parts of each model. We report our results in the following table :
> > >
> > > |      |  Ours | UV-CDN | PhyDNet | PredRNN |
> > > |:----:|:-----:|:------:|:-------:|:-------:|
> > > | PSNR | 26.27 |  25.77 |  24.26  |  34.40  |
> > >
> > >
> > > We optimized the methods and explored different settings and hyper-parameters. Best results were achieved by freezing the weights of the last layer of the PhyCell of PhyDNet, and the dynamics encoder of UV-CDN. For PredRNN, we freezed the last layer of the ST-LSTM. Our method significantly outperforms UV-CDN and PhyDNet, following the statement we made in simulation. We refer to the discussions in the paper on the simulation experiments, but in a nutshell, our method leverages several contributions, among which are the shape coefficients and the high-dimensional encoding of the dynamics encoder CoDy. PredRNN is again a very interesting baseline, since the PSNR appears to be very high. Nonetheless, qualitative assessment of the video confirms that this high score is mainly due to accurate background reconstruction. Cube trajectories are not accurately predicted, in contrast to our model. PredRNN consistently outputs a video of a non-moving stable tower for each 116 test examples. We provide visual examples in the github repository of our project : https://github.com/filteredcophy/FilteredCoPhy/tree/main/BlocktowerIRL

---

> > > > ### Comment · Reviewer_v4mS · 2021-11-30
> > > > **Increased my rating**
> > > >
> > > > Sincerely thank you for the quantitative comparisons and detailed explanations, and the visual examples are quite impressive.
> > > >
> > > > I increased my rating by 3.

---

### Author Response · Authors · 2021-11-23
**Common Answer**

We thank the reviewers for their efforts and their reviews of high quality. We are happy that they appreciated:
- **Our valuable contribution**: “This paper studies an interesting problem of counterfactual video prediction”, “... a crucial one to get counterfactual predictions”, “Introduces a sophisticated approach that is well-described and justified”, “feel that the paper is a strong contribution towards visual counterfactual reasoning“.
- **The attention given to the generation of the dataset**: “it improves the CoPhy benchmark”, “The new benchmark proposal considering Identifiability and Counterfactuality constraints is systematic and alleviates the issues in the CoPhy benchmark”, “Introduces and justifies an important new benchmark task”.
- **The quality of experiments and ablations**:  “Strong empirical evaluation that includes ablation studies and exploration of the learned representations”, “The paper provides good empirical results and extensive ablation studies for the effectiveness of each model component.”, “ It uses sensible benchmarks including previous methods and common-sense baselines”

We are also happy that one of the reviewers appreciated the large quantity of supplementary information in the appendix and even found time to look at our code.

We answered each reviewer separately using the open-review discussion tool. In a nutshell, we principally addressed the concerns spotted around the comparison baselines and upper bounds. Counterfactual reasoning on video implies to rethink the way we design dataset and models, thus finding actually comparable methods is a challenging task. We also evaluated our model against real world datasets, but fair comparisons are difficult or impossible, since these tasks are not based on counterfactual physics or rigid multibody dynamics and therefore do not correspond to the objectives of our paper.

We also improved the definition of the localized L-PSNR measured introduced in section 5, making it more precise: instead of computing the PSNR on small areas near the objects, we compute a segmentation image that separates the foreground from the background. We then compute the PSNR on the pixels belonging to the foreground. This gives a better representation of the true quality of the compared methods.

Here is a summary of the changes in the paper:
In the main paper, we changed tables 1 and 3 and figure 6.
We added the following sections to the appendix:
- Appendix E: additional evaluations and baselines on the proposed benchmark (Table 11 and Figure 16)
- Appendix F: evaluations of the proposed method on a real world dataset (Figure 17)
- Appendix G: We updated figures 18, 19 and 20 with visualization computed with PredRNN.

---

### Decision · Program_Chairs · 2022-01-20

**Decision:**

Accept (Oral)

**Comment:**

This paper introduces the Filtered-CoPhy method, an approach for learning counterfactual reasoning of physical processes in pixel space. The approach enables forecasting raw videos over long horizons, without requiring strong supervision, e.g. object positions or scene properties.

The paper initially received one strong accept, one weak accept, and one weak reject recommendations. The main reviewers' concerns relate to clarifications and consolidations in experiments, including stronger baselines, experiments on real data, or more diversity on the datasets. The rebuttal did a good job in answering reviewers' concerns, especially by providing new experimental results and analysis. Eventually, all reviewers recommended a clear acceptance after authors' feedback.

The AC's own readings confirmed the reviewers' recommendations. The proposed approach is a meaningful extension of CoPhy for the unsupervised prediction at the pixel level. The proposed approach is solid, clearly described, and overcomes important limitations of previous methods. The dataset is also an important outcome for the community. Causality and counterfactual reasoning are of primary importance for the design of effective and explainable AI prediction models: this paper brings therefore an important contribution to the ICLR community.